# Equilibrium Computation in Multidimensional Congestion Games: CSP and Learning Dynamics Approaches

**Mohammad T. Irfan**[1]        **Hau Chan**[2]        **Jared Soundy**[2]

[1]Department of Computer Science, Bowdoin College, Brunswick, Maine, USA
[2]School of Computing, University of Nebraska-Lincoln, Lincoln, Nebraska, USA

## Abstract

We present algorithms of two flavors—one rooted in constraint satisfaction problems (CSPs) and the other in learning dynamics—to compute pure-strategy Nash equilibrium (PSNE) in $k$-dimensional congestion games ($k$-DCGs) and their variants. The two algorithmic approaches are driven by whether or not a PSNE is guaranteed to exist. We first show that deciding the existence of a PSNE in a $k$-DCG is NP-complete even when players have binary and unit demand vectors. For general cost functions (potentially non-monotonic), we devise a new CSP-inspired algorithmic framework for PSNE computation, leading to algorithms that run in polynomial time under certain assumptions while offering exponential savings over standard CSP algorithms. We further refine these algorithms for variants of $k$-DCGs. Our experiments demonstrate the effectiveness of this new CSP framework for hard, non-monotonic $k$-DCGs. We then provide learning dynamics-based PSNE computation algorithms for linear and exponential cost functions. These algorithms run in polynomial time under certain assumptions. For general cost, we give a learning dynamics algorithm for an $(\alpha, \beta)$-approximate PSNE (for certain $\alpha$ and $\beta$). Lastly, we also devise polynomial-time algorithms for structured demands and cost functions.

## 1 INTRODUCTION

In non-cooperative games, a player's payoff depends on their own choice of action and the choices of actions by the other players. In general, the payoff may change depending on *who* chose a particular action. In a seminal paper, Rosenthal presented a special class of games—to become famously known as *congestion games* later—where the number of

players rather than the identities of the players choosing an action is relevant [Rosenthal, 1973]. In a congestion game, there is a set of resources (e.g., edges in a road network). Each player has a set of strategies, where each strategy is a subset of resources (e.g., paths in a network). A strategy profile consists of a strategy for each player. The cost of a resource (e.g., edge) is a function of the number of players using that resource. Given a strategy profile, a player's cost is the sum of the costs of the resources used by the player. A strategy profile is a pure-strategy Nash equilibrium (PSNE) if no player has an incentive to deviate unilaterally.

The congestion games literature can be divided into three main frontiers: unweighted, weighted one-dimensional, and weighted multidimensional. Unweighted congestion games are the classical ones [Rosenthal, 1973], where the guaranteed existence of a PSNE naturally leads to computational questions. In their seminal work, Monderer and Shapley [1996] showed that any unweighted congestion game is a *potential game*, which is appealing for learning dynamics. For unweighted congestion games on networks, if the game is *symmetric* (same start-end pair for all), then there exists a polynomial-time network-flow algorithm to find a PSNE; otherwise, the problem is PLS-complete [Fabrikant et al., 2004], even for linear cost [Ackermann et al., 2008].

In a weighted congestion game, each player has a weight or demand, and the cost of a resource is a function of the sum of the demands of the players using that resource. Unlike unweighted congestion games, a PSNE is not guaranteed to exist in weighted congestion games [Libman and Orda, 1997, Fotakis et al., 2005]. Dunkel and Schulz [2008] went one step further and showed that PSNE existence in weighted congestion games is strongly NP-complete, even for a constant number of players.

On the positive side, a PSNE is guaranteed to exist in a weighted congestion game when the cost function is linear [Fotakis et al., 2005] or exponential [Harks and Klimm, 2012]. Harks et al. [2011] characterized the existence of potential functions. Special cases involving parallel edges

Table 1: Our main results on $k$-dimensional congestion games ($k$-DCGs), $k$-class congestion games ($k$-CCGs), and variants. Notation: NPC $\equiv$ NP-Complete, $n$ = # players, $m$ = # resources, $p$ = max # strategies, $\mathbf{d}_i$ = player $i$'s demand vector, $\mathbf{d}_N = \sum_i \mathbf{d}_i$, $w_{\max} = \max_j \mathbf{d}_{N_j}$, $\check{n}$ = max # players selecting a resource in a binary $k$-DCG, or max # players of a type in a $k$-DCG with player types, $l(i)$ = nonzero-element index in $\mathbf{d}_i$ for $k$-CCG, $a_{\max}$, $b_{\max}$, and $\mathbf{z}$ are cost parameters. † We give approximation algorithms for $(\alpha, \beta)$-PSNE, which always exists.   ‡ Klimm and Schütz [2022].

| | Problem | PSNE | Time Complexity to Determine or Compute PNSE |
|---|---|---|---|
| **CSP framework** | General Cost $k$-DCG | NPC† | $\mathcal{O}\left((w_{\max})^{km}(nkp^2m^2 + nkmp(w_{\max})^{km})\right)$ |
| | Subclass: Binary $k$-DCG | NPC | $\mathcal{O}\left(\check{n}^{km}(nkp^2m^2 + \min\{nkmp\check{n}^{km}, n^{km+1}p\})\right)$ |
| | Subclass: $k$-CCG | NPC | $\mathcal{O}\left((w_{\max})^{km}(np^2m^2 + nkpm(w_{\max})^m)\right)$ |
| | Subclass: $k$-DCG with player types | NPC | $\mathcal{O}\left((\check{n})^{\tau m}(np^2m^2 + n\tau pm(\check{n})^m) + \tau nk\right)$ |
| **Learning dynamics** | Linear Cost $k$-DCG | Always‡ | $\mathcal{O}\left(nkpm^2 \times n^2m(a_{\max} + b_{\max})\frac{\max_i[\mathbf{z}\cdot\mathbf{d}_i]^2}{\min_i[\mathbf{z}\cdot\mathbf{d}_i]}\right)$ |
| | Linear Subclass: Binary $k$-DCG | Always | $\mathcal{O}\left(nkpm^2 \times n^2m(a_{\max} + b_{\max})\left(k\max_j z_j\right)^2\right)$ |
| | Linear Subclass: $k$-CCG | Always | $\mathcal{O}\left(nkpm^2 \times n^2m(a_{\max} + b_{\max})\frac{\max_j z_j^2}{\min_j z_j}\frac{\max_i d_{i,l(i)}^2}{\min_i d_{i,l(i)}}\right)$ |
| | Exponential Cost $k$-DCG | Always‡ | $\mathcal{O}\left(nkpm^2 \times \frac{e}{e-1}\left(m\exp(\mathbf{z}\cdot\mathbf{d}_N)a_{\max} + nmb_{\max}\right)\right)$ |
| **Structured** | Ordered $\mathbf{d}_i$'s, nondec. cost, singleton strt. | Always | $\mathcal{O}(n\log n + nmk)$ |
| | Ordered $\mathbf{d}_i$'s, nondec. cost, shared strt. | Always | $\mathcal{O}(n\log n + npmk)$ |
| | Structured cost, singleton strt. | Always | $\mathcal{O}(n\log n + nmk)$ |

have also received attention [Milchtaich, 1996, Fotakis et al., 2002, Gairing et al., 2004, Mavronicolas et al., 2007].

Multidimensional congestion games are a very recent frontier which we investigate here. Introduced by Klimm and Schütz [2014], this class of games is a generalization of weighted congestion games where the demand of each player is a $k$-dimensional vector. Very recently, Klimm and Schütz [2022] have shown that certain affine and exponential cost functions are the only ones for which a PSNE exists for sure. Their characterization leads to the following *computational* questions investigated here: *How can we compute a PSNE (if it exists) in multidimensional congestion games and their variants? How hard is this computation?*

These questions are motivated by many real-world applications. Advances in multidimensional congestion games may contribute to richer traffic models that account for the heterogeneity in vehicles (e.g., weight, length, etc.). Such multidimensional models were envisioned by transportation researchers many decades ago [Dafermos, 1972] and are now topics of active investigation [Van Lint et al., 2008, Pi et al., 2019, Wang et al., 2019]. Computational advances may also contribute to various other application areas—wireless networks [Yamamoto, 2015], distributed systems [Nadig et al., 2022, 2019], telecommunication [Altman et al., 2006], and smart grids [Fadlullah et al., 2011], to name just a few.

**Our Contributions**

Driven by whether or not a PSNE is guaranteed to exist, we take two fundamentally different computational approaches

inspired by CSPs and learning dynamics. The CSP approach can handle *any* $k$-DCGs (for which a PSNE may not exist), whereas learning dynamics can handle certain $k$-DCGs with a PSNE. We exploit the structure of multidimensional congestion games to give new computational insights into $k$-dimensional congestion games ($k$-DCGs) and their variants, as summarized in Table 1.

For general $k$-DCGs, we devise a CSP whose dual decouples the players' strategies. We give algorithms that utilize this decoupling and run in polynomial time under certain assumptions (Section 4). [1] *To our knowledge, this CSP framework is new within the rich congestion games literature spanning over five decades.* The significance of our CSP framework lies not only in the exponential savings it offers compared to well-known CSP algorithms but also in its applicability beyond congestion games.

For linear and exponential cost, we give iterative learning dynamics algorithms for $k$-DCGs and their variants by deriving and bounding weighted potential functions based on the structure of the game (Section 5). For general cost, we show that for certain $\alpha$ and $\beta$, there is always an $(\alpha, \beta)$-approximate PSNE that can be computed via learning dynamics. We also give polynomial-time algorithms for structured costs and demands (Section 6).

The significance of our computational results can be best understood against the backdrop of hardness results (Section 3). We show that deciding the existence of a PSNE in a $k$-

---

[1] Polynomial-time algorithms are unlikely due to PLS-completeness results for unweighted network congestion games with linear cost functions [Ackermann et al., 2008].

DCG is NP-complete even for binary demand vectors (and other special cases). Put together, this paper addresses computational questions while giving new insights for provably hard problems on congestion games.

## 2   PRELIMINARIES

We formally define multi-dimensional congestion games and related game-theoretic terms. Roughly speaking, a multi-dimensional congestion game is a natural generalization of weighted congestion games where the weight or demand of each player is a multidimensional vector. The cost of each resource is a function of the aggregated demands of the players using that resource.

More formally, a $k$-dimensional congestion game ($k$-DCG) consists of a set $N = \{1, \ldots, n\}$ of $n$ players and a set $R = \{1, \ldots, m\}$ of $m$ resources. Each player $i \in N$ has two elements: (1) a strategy set $S_i \subseteq 2^R \setminus \{\emptyset\}$, defined to be subsets of resources that $i$ can select and (2) a $k$-dimensional demand vector $\mathbf{d}_i = (d_{i_1}, \ldots, d_{i_k}) \in \mathbb{R}^k$, consisting of the weight or demand of player $i$ at each dimension $1, \ldots, k$. Each resource $r \in R$ has a cost function $c_r : \mathbb{R}^k \to \mathbb{R}$ that maps $k$-dimensional real-valued vectors to real numbers. We use $p = \max_{i \in N} |S_i|$ to denote the maximum number of strategies for any player. We make the standard assumption that demands are non-negative integer vectors [Panagopoulou and Spirakis, 2007, Dunkel and Schulz, 2008, Christodoulou et al., 2023].

Given a strategy profile $\mathbf{s} = (s_1, \ldots, s_n) \in S = S_1 \times \ldots \times S_n$ of $n$ players, let $\mathbf{x}_r(\mathbf{s}) = \sum_{i \in N : r \in s_i} \mathbf{d}_i$ be the aggregated $k$-dimensional demand vector of the players who select resource $r$ under the strategy profile $\mathbf{s} \in S$. Naturally, given a strategy profile $\mathbf{s}$, the cost function of player $i$ is defined to be $\pi_i(\mathbf{s}) = \pi_i(s_i, \mathbf{s}_{-i}) = \sum_{r \in s_i} c_r(\mathbf{x}_r(\mathbf{s}))$, i.e., the sum of the costs of the resources selected by player $i$ under $s_i$, given others' strategies $\mathbf{s}_{-i}$.

We are interested in computing PSNE in $k$-DCGs and their variants listed below. We present these variants with motivating examples from the domain of load balancing in distributed systems [Nadig et al., 2022, 2019, Anantha et al., 2017]. $k$-DCGs naturally model various dimensions of user demands in distributed systems, such as bit rates, latency, error tolerance, and throughputs.

- $k$-DCGs with binary demand vectors $\mathbf{d}_i \in \{0, 1\}^k$ $\forall i$. Example: data flow in distributed systems can be short-lived or long-lived, bursty or deterministic, etc.

- $k$-class congestion games ($k$-CCGs), where each demand vector has one positive element, the rest being zeros. Example: different use-cases (each with its own traffic pattern), such as streaming, video conferencing, web browsing, etc.

- $k$-DCGs with player types, where players of the same

type are characterized by the same demand vector. Example: categories of traffic on a campus network: VPN, student access, scientific computation, etc.

We next define PSNE and approximate PSNE– two solution concepts of our interest.

**Definition 1.** *(Pure-Strategy Nash Equilibrium (PSNE)) A strategy profile* $\mathbf{s}^* = (s_1^*, \ldots, s_n^*) \in S$ *is a pure-strategy Nash equilibrium (PSNE) in a $k$-DCG if and only if for each player* $i \in N$ *and any* $s_i' \in S_i$, *we have that* $\pi_i(\mathbf{s}^*) \leq \pi_i(s_i', \mathbf{s}_{-i}^*)$.

**Definition 2.** *(($\alpha, \beta$)-PSNE) A strategy profile* $\mathbf{s} = (s_1, \ldots, s_n) \in S$ *is an ($\alpha, \beta$)-approximate PSNE in a $k$-DCG for some* $\alpha \geq 1$ *and* $\beta \geq 0$ *if and only if for each player* $i \in N$ *and any* $s_i' \in S_i$, *we have that* $\pi_i(\mathbf{s}) \leq \alpha \pi_i(s_i', \mathbf{s}_{-i}) + \beta$. *When we mention $\alpha$-PSNE (without $\beta$), we mean* $\beta = 0$.

### Constraint Satisfaction Problem (CSP)

A CSP is specified by a set of *variables*, a *domain* for each variable, and a set of *constraints*, each constraint being over a subset of variables known as its *scope*. A CSP asks us to assign a value to each variable from their respective domains so that all the constraints are satisfied. A wide range of problems, such as Boolean satisfiability, map coloring, scheduling, and even PSNE computation in games, can be modeled as CSPs [Dechter, 2003, Gottlob et al., 2003].

We often represent the structural information of a (primal) CSP using a *primal constraint network*, where each node represents a variable, and each edge connects two variables that appear together in a constraint (potentially with other variables). As a result, unless the constraints are binary, we cannot identify the scope of a constraint just by looking at the primal constraint network.

A CSP also has a *dual constraint network*, where each variable represents a constraint, and each edge connects two constraints with shared variables in their scopes and is labeled with these shared variables. The dual constraint network leads to the *dual CSP*, where the domain of each dual variable is computed as follows: Consider its corresponding primal constraint and assign values to the scope of the primal constraint to satisfy it. Such assignments constitute the domain of the dual variable. Furthermore, the dual CSP enforces the edge-wise dual constraint that each primal variable shared between any two dual variables must have the same value in both. Therefore, the dual CSP is a reformulation of the primal CSP and contains only binary constraints.

## 3   COMPUTATIONAL COMPLEXITY

We show that deciding the existence of a PSNE in special variants of $k$-DCGs is NP-complete. The NP-hardness of

general $k$-DCGs is not surprising because determining a PSNE in weighted congestion games (i.e., when $k = 1$) is already strongly NP-complete [Dunkel and Schulz, 2008].

What is surprising is that we show that determining the existence of a PSNE in $k$-DCGs is NP-complete even when each player $i$'s $k$-dimensional demand vector $\mathbf{d}_i$ is a binary vector (even a unit vector) for some polynomially bounded $k$. In sharp contrast, there is always a PSNE in unweighted (1-dimensional) congestion games [Rosenthal, 1973]. Furthermore, if the players have the same demand vector, the game is guaranteed to have a PSNE by reducing it to an unweighted congestion game. We have the following result.

**Theorem 3.** *Deciding the existence of a PSNE in a $k$-DCG is NP-complete even when the demand vector $\mathbf{d}_i$ of each player $i \in N$ is a binary vector and $k$ is sublinear in the number of players. That is, $\mathbf{d}_i \in \{0, 1\}^k$ for all $i$ and $k = \mathcal{O}(\log n)$.*

*Proof Sketch.* The problem is in NP because verifying that a strategy profile $\mathbf{s}^* \in S$ is a PSNE takes polynomial time. For NP-hardness, we reduce from weighted congestion games [Dunkel and Schulz, 2008]. Given a weighted congestion game we construct a $k$-DCG with identical sets of players, resources, and actions. In the $k$-DCG game we give the players binary demand vectors equivalent to the binary representations of the integer weights from the weighted congestion game. The length of the demand vector is set to $k = \lfloor \log \max_{i \in N} \widetilde{d}_i \rfloor + 1$ where $\widetilde{d}_i$ is the integer weight of player $i$ in the weighted congestion game. Finally, we construct cost functions for $k$-DCGs that we show to yield the same cost given the same strategy profile for all players. Therefore, a strategy profile is a PSNE in one game if and only if it is a PSNE in the other game. $\square$

Next, we investigate whether PSNE computation is easier for restricted demands. Unfortunately, even when the binary demand vector is a unit vector, the problem remains hard.

**Theorem 4.** *Deciding the existence of a PSNE in a $k$-DCG (or a $k$-CCG) is NP-complete even when the demand vector $\mathbf{d}_i$ of each player $i \in N$ is a binary unit vector and $k$ is linear of the number of players. That is, $\mathbf{d}_i \in \{\mathbf{x} \in \{0, 1\}^k; \sum_{j=1}^{k} x_j = 1\}$ for all $i$ and $k = \mathcal{O}(n)$.*

*Proof Sketch.* The problem is clearly in NP. The NP-hardness reduction is from weighted congestion games. $\square$

## 4 GENERAL COST: A CSP APPROACH

We can formulate the PSNE computation problem in a $k$-DCG as a CSP, which consists of (1) a variable for each player, (2) the domain of a variable being the corresponding player's strategy set, and (3) a best-response constraint for each player $i$, representing $i$'s best responses $s_i$ to any $\mathbf{s}_{-i}$.

As illustrated in Fig. 1 (a) and (b), the nature of the $n$-ary best-response constraints means that both the primal and the dual constraint networks are complete networks. Furthermore, *all* players appear on each edge of the dual network. This portrays a grim picture because it is hard to design efficient algorithms without decoupling the players' strategies. For example, one solution approach is to check each strategy profile for a PSNE by verifying Definition 1. Letting $p = \max_{i \in N} |S_i|$, this approach takes $\mathcal{O}(np^{n+1})$ time, which is exponential in the number of players.

The grave computational implication of not decoupling the players' strategies leads us to a key technical insight. Instead of using the above CSP, we first construct a different CSP for $k$-DCGs and then consider its dual. In the new CSP, the variables are the players and the *configuration* $Y$ of the game. The domain of each player $i$ is their strategy set $S_i$ and that of $Y$ is the set of all $k$-dimensional aggregated demand vectors for $m$ resources, $\mathbf{y} \equiv (\mathbf{y}_1, \mathbf{y}_2, ..., \mathbf{y}_m)$. There are $n$ binary constraints, each capturing a player's *best response to a configuration*. We use the structure of $k$-DCGs to define such best responses: For any configuration $\mathbf{y}$, a player $i$'s best-response strategies are $s_i \in S_i$ that minimize the cost $\sum_{r \in s_i} c_r(\mathbf{y}_r)$. There is an additional *feasibility constraint* that enforces that the strategy profile $\mathbf{s}$ assigned to the players leads to the aggregated demand vectors $(\mathbf{y}_1, \mathbf{y}_2, ..., \mathbf{y}_m)$ assigned to $Y$; i.e., $\mathbf{x}_1(\mathbf{s}) = \mathbf{y}_1, \mathbf{x}_2(\mathbf{s}) = \mathbf{y}_2, ..., \mathbf{x}_m(\mathbf{s}) = \mathbf{y}_m$. An example of the primal constraint network for this CSP is shown in Fig. 1(c) and its dual in Fig. 1(d). Most notably, as elaborated in the next paragraph, the dual CSP allows us to decouple the players' strategies from each other.

*To our knowledge, this dual CSP, which grounds our algorithmic framework, has not been studied in the congestion games literature before.* To formalize this dual CSP, each dual node is a primal constraint. So, there is a dual node $v_{i,Y}$ for each player $i$'s best response to the configuration variable $Y$, and there is one dual variable $v_{N,Y}$ for the feasibility constraint making sure that the strategies assigned to the players lead to the aggregated demands assigned to $Y$ (see Fig. 1(d)). For each $i \in N$, there is an edge between $v_{N,Y}$ and $v_{i,Y}$ labeled with the shared variables $i, Y$. For any $i \neq j \in N$, there is an edge between $v_{i,Y}$ and $v_{j,Y}$ labeled with the shared variable $Y$. Unlike the straightforward dual (Fig. 1(b)), this new dual (Fig. 1(d)) decouples the players' strategies by virtue of not having all the players appear together on any edge.

We devise algorithms based on this dual CSP. As described in Section 2, each dual variable has a domain consisting of satisfying assignments for the corresponding primal constraint, and the edges in the dual constraint network lead to dual constraints that ensure that the shared primal vari-

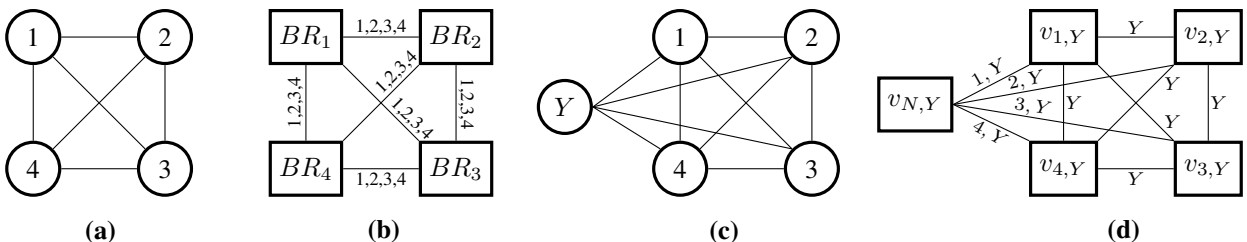

Figure 1: The four constraint networks shown here illustrate our key technical insight.
**(a)** A typical CSP (not used here) for $N = \{1, \cdots, 4\}$: Each node is a player with their strategy set as the domain, and each player has a best-response constraint involving all the players.
**(b)** The dual of **(a)**: Each dual node $BR_i$ represents player $i$'s best-response constraint with its domain being strategy profiles $(s_i, \mathbf{s}_{-i})$ where $s_i$ is $i$'s best response to $\mathbf{s}_{-i}$. Each edge shows that *all* players are shared between its endpoints, which makes it hard to decouple the strategies of the players.
**(c)** A new CSP we present where in addition to the players, there is a node $Y$ for the configuration representing aggregated demand vectors $(\mathbf{y}_1, \mathbf{y}_2, ..., \mathbf{y}_m)$. The constraints are: (1) *best-response constraint:* Each player $i$ plays its best response to $Y$, leading to the edge $(i, Y)$, and (2) *feasibility constraint:* the strategies assigned to the players are consistent with the configuration assigned to $Y$, leading to all the edges because this constraint involves all the nodes.
**(d)** The dual of **(c)**: Each $v_{i,Y}$ node (for $i = 1, \cdots, 4$) represents the best response constraint described above, and the $v_{N,Y}$ node represents the feasibility constraint described above. Each edge is labeled with the shared variables between its endpoints. Most notably, the edges show the decoupling of the players' strategies. Contrast this with **(b)**.

ables across any edge are assigned the same value in both endpoints of the edge.

Before presenting our algorithmic framework, we show that the dual CSP has a solution if and only if there is a PSNE. To see why, note that the assignments $(s_i, \mathbf{y})$ made to the $v_{i,Y}$ variables capture the players' best responses to $\mathbf{y}$, and the edge label between any two $v_{i,Y}$ and $v_{j,Y}$ variables enforces sharing the same $\mathbf{y}$ in these assignments. Furthermore, the assignment $(\mathbf{s}, \mathbf{y}')$ made to the $v_{N,Y}$ variable makes sure that the strategy profile $\mathbf{s}$ leads to the configuration $\mathbf{y}'$, and the labels on the edges connecting $v_{N,Y}$ to $v_{i,Y}$ enforce that $\mathbf{s} = (s_1, \cdots, s_n)$ and $\mathbf{y} = \mathbf{y}'$.

Our algorithmic framework consists of two procedures. Procedure 1 computes the domains of each $v_{i,Y}$ dual variable and Procedure 2 searches for a solution using the computed domains. As a preview, our algorithms are polynomial in $n$ (the number of players), $p$ (the maximum number of strategies for any player), and a maximum weight term when $k$ (number of dimensions) and $m$ (number of resources) are bounded. This is useful when the number of resources and strategies is constant but the number of players can be large. In fact, even with a constant number of players, determining PSNE existence in a weighted congestion game is already strongly NP-complete [Dunkel and Schulz, 2008].

As Fig 1(d) shows, $Y$ is shared across all edges. Therefore, we parameterize our algorithms by any configuration given as input. This leads to the question of how many configurations there can be. The demand vector $\mathbf{d}_i = (d_{i_1}, ..., d_{i_k})$ of each player $i$ being an integer vector (standard assumption [Dunkel and Schulz, 2008]), we define $w_j = \sum_{i \in N} d_{i_j}$ for each $j = 1, ..., k$. Letting $w_{\max} = \max_{j \in [k]} w_j$, we

have $\mathbf{y}_1, \mathbf{y}_2, ..., \mathbf{y}_m \in \{0, ..., w_{\max}\}^k$. Thus, we only need to consider at most $(w_{\max} + 1)^{km}$ or $\mathcal{O}((w_{\max})^{km})$ configurations. We are now ready for the algorithms.

**Procedure 1: Compute Domains of Dual Variables $v_{i,Y}$**

Given a configuration $\mathbf{y} \equiv (\mathbf{y}_1, \mathbf{y}_2, ..., \mathbf{y}_m)$, where each $\mathbf{y}_j \in \{0, ..., w_{\max}\}^k$, we compute the set of strategies for each player $i$ that makes $i$ "happy" under the configuration. To do this, abusing the notation $\pi_i$ slightly, we define and compute, for any $i \in N$, $s_i, s_i' \in S_i$, and $s_i \neq s_i'$,

$$\pi_i(s_i, \mathbf{y}) = \sum_{r \in s_i} c_r(\mathbf{y}_r)$$

$$\pi_i(s_i, \mathbf{y}, s_i') = \sum_{r \in s_i' \cap s_i} c_r(\mathbf{y}_r) + \sum_{r \in s_i' \setminus s_i} c_r(\mathbf{y}_r + \mathbf{d}_i)$$

$$BR_i(\mathbf{y}) = \{s_i \in S_i \mid \forall s_i' \in S_i, \pi_i(s_i, \mathbf{y}) \leq \pi_i(s_i, \mathbf{y}, s_i')\}$$

The first equation calculates player $i$'s cost. The second calculates player $i$'s cost when deviating from $s_i$ (under $\mathbf{y}$) to $s_i'$. $BR_i(\mathbf{y})$ computed in the last equation is the set of $i$'s best responses to $\mathbf{y}$. Therefore, the domain of $v_{i,Y}$ is the union of sets $\{(s_i, \mathbf{y}) \mid s_i \in BR_i(\mathbf{y})\}$ for all $\mathbf{y}$.

We deliberately do not compute the domain of $v_{N,Y}$ (the dual variable for the primal feasibility constraint) because it may contain numerous strategy profiles that are not PSNE. We next show in Procedure 2 how we can search for a PSNE without explicitly computing the domain of $v_{N,Y}$.

## Procedure 2: Search for PSNE

Given a configuration $\mathbf{y} \equiv (\mathbf{y}_1, \mathbf{y}_2, ..., \mathbf{y}_m)$, a PSNE under it is a strategy profile $\mathbf{s} = (s_1, ..., s_n)$ such that (1) $(s_i, \mathbf{y})$ is in the domain of $v_{i,Y}$ for each player $i$, and (2) $\mathbf{x}_1(\mathbf{s}) = \mathbf{y}_1, \mathbf{x}_2(\mathbf{s}) = \mathbf{y}_2, ..., \mathbf{x}_m(\mathbf{s}) = \mathbf{y}_m$. The first condition enforces players' best responses to $\mathbf{y}$, while the second condition enforces the feasibility constraint. We get the following general result.

**Theorem 5.** *For any $k$-DCG, there is an algorithm to determine the existence of a PSNE in $\mathcal{O}((w_{\max})^{km}(nkp^2m^2 + nkmp(w_{\max})^{km}))$. The algorithm is polynomial in $n$, $p$, and $w_{\max}$, when $m$ and $k$ are constants.*

*Proof Sketch.* Procedure 1 runs in $\mathcal{O}(nkp^2m^2)$ for all players. Procedure 2 can be done efficiently using dynamic programming (DP), where we (1) first order the players $1, ..., n$ and (2) create a binary table $T_i(\mathbf{y}_1', \mathbf{y}_2', ..., \mathbf{y}_m') \in \{0, 1\}$ for each $\mathbf{y}_1', \mathbf{y}_2', ..., \mathbf{y}_m' \in \{0, ..., w_{\max}\}^k$ of size $\mathcal{O}((w_{\max})^{km})$ for each player $i$. We first initialize $T_0(\mathbf{0}, ..., \mathbf{0}) = 1$ where we have an all zero configuration. We then define $T_i(\mathbf{y}_1', \mathbf{y}_2', ..., \mathbf{y}_m') = 1$ if and only if there is $\overline{\mathbf{y}}_1, \overline{\mathbf{y}}_2, ..., \overline{\mathbf{y}}_m$ such that $T_{i-1}(\overline{\mathbf{y}}_1, \overline{\mathbf{y}}_2, ..., \overline{\mathbf{y}}_m) = 1$ and for some $s_i \in BR_i(\mathbf{y}_1, \mathbf{y}_2, ..., \mathbf{y}_m)$, $\mathbf{y}_r' = \overline{\mathbf{y}}_r + \mathbb{1}[r \in s_i]\mathbf{d}_i$ for each $r \in R$. Table $T_i$ can be constructed by looking at all the 1 entries of $T_{i-1}$ and adding the player demand vector to the corresponding resources for each $s_i \in BR_i(\mathbf{y}_1, \mathbf{y}_2, ..., \mathbf{y}_m)$. The DP runs in $\mathcal{O}(nkmp(w_{\max})^{km})$. $\qquad\square$

Algorithm 1 presents the decision version of the DP algorithm given in the proof of Theorem 5. Please note that we are going to refine it for variants of $k$-DCGs.

To see the significance of the above result, note that we can use well-known algorithms to solve the dual CSP. For instance, backtracking algorithms with graph-based learning can solve the dual CSP in $\mathcal{O}\left((n+1)^2 \cdot \left(2 \cdot p^n \cdot w_{\max}^{km}\right)^{n+1}\right)$ time, which is exponential in $n$ [Dechter, 2003][Ch 6]. In contrast, our algorithm guarantees an exponential saving.

### VARIANT: BINARY DEMAND VECTORS

In Section 3, we showed that $k$-DCGs with binary demand vectors are provably hard. We can still apply Theorem 5 to derive a pseudopolynomial time algorithm when $k$ and $m$ are bounded. However, an improved analysis gives us the following result. Note that in the case of binary demand vectors, any $j$-th element of an aggregated demand vector corresponds to the number of players having the $j$-th bit of their demand vector "on." Therefore, for clarity, we use $\check{n} = \max_{j \in [k]} \sum_{i \in N} d_{i_j}$ in place of $w_{\max}$ to denote the maximum number of players having a demand vector bit on. The following result is particularly interesting when $\check{n} \ll n$.

---

**Algorithm 1:** Determine if there is a PSNE

**Input:** A multidimensional congestion game
**Output:** TRUE if a PSNE exists, FALSE otherwise.

1 **for** *configuration* $\mathbf{y}_1, \mathbf{y}_2, \cdots, \mathbf{y}_m \in \{0, \cdots, w_{\max}\}^k$ **do**
2      **for** *each player* $i \in \{1, 2, \cdots, n\}$ **do**
3          Compute $BR_i(\mathbf{y}_1, \mathbf{y}_2, \cdots, \mathbf{y}_m)$
4      **end**
5      Create a binary table $T_0$ with $T_0(\mathbf{0}, \cdots, \mathbf{0}) = 1$
6      **for** *each player* $i \in \{1, 2, \cdots, n\}$ **do**
7          Create a binary table $T_i$ as follows:
8          **for** *each* $\overline{\mathbf{y}}_1, \overline{\mathbf{y}}_2, \cdots, \overline{\mathbf{y}}_m$ *such that* $T_{i-1}(\overline{\mathbf{y}}_1, \overline{\mathbf{y}}_2, \cdots, \overline{\mathbf{y}}_m) = 1$ **do**
9              **for** $s_i \in BR_i(\mathbf{y}_1, \mathbf{y}_2, \cdots, \mathbf{y}_m)$ **do**
10                 Set $T_i(\mathbf{y}_1', \mathbf{y}_2', \cdots, \mathbf{y}_m') = 1$ where $\mathbf{y}_r' = \overline{\mathbf{y}}_r + \mathbb{1}[r \in s_i]\mathbf{d}_i$ for each $r \in R$
11              **end**
12          **end**
13      **end**
14      **if** $T_n(\mathbf{y}_1, \mathbf{y}_2, \cdots, \mathbf{y}_m) = 1$ **then**
15          return TRUE
16      **end**
17 **end**
18 return FALSE

---

**Theorem 6.** *For $k$-DCGs with binary demand, there is an $\mathcal{O}(\check{n}^{km}(nkp^2m^2 + \min\{nkmp\check{n}^{km}, n^{km+1}p\}))$-time algorithm to compute a PSNE or decide none exists. The algorithm is polynomial in $n$ and $p$ when $m$ and $k$ are constants.*

*Proof Sketch.* Putting $w_{\max} = \check{n}$ in Theorem 5, the running time is $\mathcal{O}(\check{n}^{km}(nkp^2m^2 + nkmp\check{n}^{km}))$. However, using a different analysis that exploits the bit-vector structure, we can shave off a factor of $km$ from the second term at the expense of having $n^{km}$ instead of $\check{n}^{km}$. This would be useful when $\check{n} \approx n$. The main idea is when we consider player $i$ in Procedure 2, the number of configurations for $T_i$ is at most $(i+1)^{km}$, leading to $\mathcal{O}\left(\sum_{i=1}^{n}\left[(i+1)^{km} + kpmi^{km}\right]\right)$ or $\mathcal{O}(n^{km+1}p)$ time for Procedure 2. $\qquad\square$

### VARIANT: $k$-CLASS CONGESTION GAME ($k$-CCG)

Let the *class of player $i$* be the index where the positive element appears in $\mathbf{d}_i$. Although Theorem 5 can be directly applied to this case, we can exploit the structure of the game to improve the running time. The key intuition is that the players can be partitioned according to their classes. The players in a class $j \in [k]$ can only affect the $j$-th index of the aggregated demand on any resource. That is, they affect the $j$-th index of each of $\mathbf{y}_1, \mathbf{y}_2, ..., \mathbf{y}_m$. As a result, Procedure 2 can be broken into $k$ different computational tasks, each

corresponding to a class. This idea leads us to the following result. *Notably, compared to Theorem 5, this partition-based algorithm removes a $k$ term from the exponent.*

**Theorem 7.** *For $k$-CCGs, there is an $\mathcal{O}((w_{\max})^{km}(np^2m^2 + nkpm(w_{\max})^m))$ algorithm to compute a PSNE or decide none exists. The algorithm is polynomial in $n$, $p$, and $w_{\max}$ when $m$ and $k$ are constants.*

*Proof Sketch.* As a preprocessing step, we partition the players into $C_1, ..., C_k$ based on their classes. We now do the following operations in each partition $C_j$ independently. We start the DP by ordering the players in $C_j$ as $1, 2, ..., |C_j|$ (wlog). We then create a binary table $T_i(z_1, z_2, ..., z_m) \in \{0, 1\}$ for each $z_1, z_2, ..., z_m \in \{0, ..., w_{\max}\}$ of size $\mathcal{O}((w_{\max})^m)$ for each player $i$ in $C_j$. We initialize $T_0(0, ..., 0) = 1$. We then define $T_i(z_1, z_2, ..., z_m) = 1$ if and only if there is $z_1', z_2', ..., z_m'$ such that $T_{i-1}(z_1', z_2', ..., z_m') = 1$ and for some $s_i \in BR_i(\mathbf{y}_1, \mathbf{y}_2, ..., \mathbf{y}_m)$, $z_r = z_r' + \mathbb{1}[r \in s_i]d_{ij}$ for each $r \in R$. We have a PSNE if and only if for each partition $C_j$, $T_{|C_j|}(y_{1j}, y_{2j}, ..., y_{mj}) = 1$. $\square$

We next consider the special case of $k$-CCGs with binary demand vectors (i.e., exactly one bit is "on" in each player's demand vector). This will be useful when we consider player types next. We get the following corollary from Theorems 6 and 7. Once again, the result is interesting when $\check{n} \ll n$.

**Corollary 8.** *For $k$-CCGs with binary demand vectors, there is an $\mathcal{O}((\check{n})^{km}(np^2m^2 + nkpm(\check{n})^m))$-time algorithm to compute a PSNE or decide none exists. The algorithm is polynomial in $n$ and $p$ when $m$ and $k$ are constants.*

## VARIANT: $k$-DCG WITH PLAYER TYPES

To motivate this variant, consider a road-traffic setting. There are different types of vehicles, and vehicles of the same type share similarities in their demand vectors. We define players to be of the same type if their demand vectors are the same. Although this setting is very natural, to our knowledge, it has not been fully explored in the literature. Here, other than player types, we do not make any assumptions about the demands or cost functions. While this variant is NP-hard (reduction from $k$-DCG by making a type for each player), the following result is very appealing when the maximum number of players of any type $\check{n} \ll n$.

**Theorem 9.** *Given a $k$-DCG with $\tau$ types of players and at most $\check{n}$ players of any type, there is an $\mathcal{O}((\check{n})^{\tau m}(np^2m^2 + n\tau pm(\check{n})^m) + \tau nk)$ time algorithm to compute a PSNE or decide that there exists none. The algorithm is polynomial in $n$ and $p$ for bounded $m$ and $\tau$.*

*Proof.* Let $(N, R, \{S_i, \mathbf{d}_i\}_{i \in N}, \{c_r\}_{r \in R}, k)$ be a $k$-DCG

instance with $\tau$ types of players. We reduce this instance to a PSNE-equivalent $\tau$-DCG instance $(N, R, \{S_i, \widetilde{\mathbf{d}}_i\}_{i \in N}, \{\widetilde{c}_r\}_{r \in R}, \tau)$ as follows. First, we partition the $k$-DCG players into $\tau$ types and store the $k$-dimensional demand vector (from $k$-DCG) of any player of type $t$ into $\overline{\mathbf{d}}_t$ (i.e., $\overline{\mathbf{d}}_t = \mathbf{d}_i$ if player $i$ is of type $t$). This takes $\mathcal{O}(\tau nk)$ time. For each player $i$ of type $t$ in $\tau$-DCG, we define a $\tau$-dimensional unit demand vector $\widetilde{\mathbf{d}}_i$ where only the $t$-th element is 1, the rest being 0. Given a $\tau$-dimensional aggregated demand vector $\widetilde{\mathbf{x}}_r(\mathbf{s}) = (\widetilde{\mathbf{x}}_r(\mathbf{s})_1, ..., \widetilde{\mathbf{x}}_r(\mathbf{s})_\tau)$, where any $t$-th element represents the total number of players of type $t$ using $r$, we define the cost function $\widetilde{c}_r(\widetilde{\mathbf{x}}_r(\mathbf{s})) = c_r\left(\sum_{t=1}^\tau (\widetilde{\mathbf{x}}_r(\mathbf{s}))_t \overline{\mathbf{d}}_t\right)$. Thus, $\widetilde{c}_r(\widetilde{\mathbf{x}}_r(\mathbf{s})) = c_r(\mathbf{x}_r(\mathbf{s}))$, where $\mathbf{x}_r(\mathbf{s})$ is the aggregated demand in the $k$-DCG instance under $\mathbf{s}$. Therefore, with the PSNE-equivalent $\tau$-DCG being a $\tau$-CCG with binary demands and $\check{n} = \max_{j \in [\tau]} \sum_{i \in N} \widetilde{d}_{ij}$, Corollary 8 gives us the result. $\square$

Comparing Theorems 9 and 5, when we have the type information, Theorem 9 offers a major saving in running time by replacing $(w_{\max})^{km}$ with $\check{n}^{\tau m}$ in the multiplicative factor as well as $(w_{\max})^{km}$ with $\check{n}^m$ (note the exponential saving of $k$) in the interior expression. These savings are especially pronounced when $\check{n}$ is small.

Theorem 9 can be extended to general $k$-DCGs *without* any player types, in which case $\check{n} = n$. This insight helps us avoid potentially large $w_{\max} \gg n$ in the running time of Theorem 5 by using Theorem 9 instead. Further running time reduction for the case of $\check{n} = n$ is possible through the alternative analysis given in the proof of Theorem 6.

## EXPERIMENTS

We have performed experiments to investigate the practical aspects of the CSP framework for non-monotonic $k$-DCGs with binary demand vectors. Even with small-scale experiments, we show that the theoretical running time greatly overestimates the practical, worst-case running time. These experiments further show that our CSP framework supports a variety of implementation possibilities.

We have implemented two instantiations of the framework: (1) Table-based DP (TDP), where we use bit vectors to implement the tables, and (2) Set-based DP (SDP), where we use hash-set data structures to represent the tables. In addition, we have implemented the brute-force (BF) algorithm mentioned for the CSP shown in Fig. 1(a). BF is the only prior algorithm known to us for general $k$-DCGs.

All three algorithms exhaustively search for all PSNE and discard a strategy profile as soon as it is clear it cannot lead to a PSNE. We have benchmarked the theoretical running time in the worst case by running Procedure 2 on a small table and extrapolating that running time to the table size

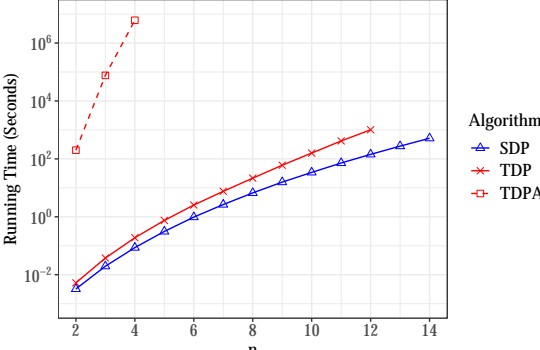

Figure 2: Running-time comparison among table-based DP (TDP), set-based DP (SDP), and table-based DP asymptotic (TDPA). Encouragingly, even at small scales, TDPA hugely overestimates the actual running time. Here, $m = 4$ and $k = 2$.

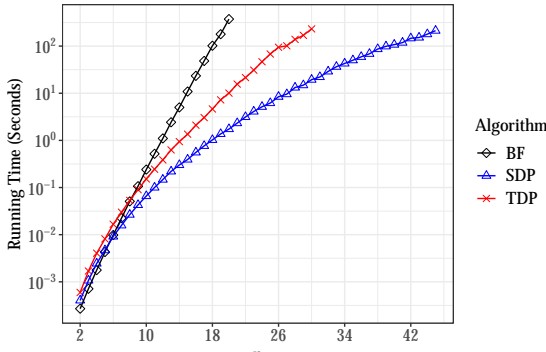

Figure 3: Running-time comparison among brute force (BF), table-based DP (TDP), and set-based DP (SDP). Even at small scales, brute force does not finish within the allocated time when $n > 20$. SDP is the fastest. Here, $m = 2$ and $k = 3$.

appearing in Theorem 5. We call this table-based DP asymptotic (TDPA). We have used non-monotonic $k$-DCGs with $m$ parallel links. Each parameter-combination was repeated 15 times. See the Appendix for details.

Fig. 2 shows that the asymptotic running time greatly overestimates the actual running time. E.g., for $n = 4$, TDPA is about eight orders of magnitude slower than SDP. Furthermore, Fig. 3 shows that SDP and TDP outperform BF easily, even for very small $n$. For example, SDP is two orders of magnitude faster than BF for $n = 18$. These signify the practical appeal of our CSP framework against the backdrop of hardness results. Most importantly, Procedure 2 opens up a range of possibilities for new CSP-based search algorithms rooted in, for example, backjumping and learning [Kumar, 1992, Dechter, 2003, Van Beek, 2006, Rossi et al., 2008], backtracking with tree decomposition [Jégou and Terrioux, 2003], AND/OR search [Marinescu and Dechter, 2009], etc. We leave a comprehensive experimental study as future work.

## 5 LEARNING DYNAMICS APPROACH

The second class of algorithms we present is grounded in learning dynamics, which often presents a natural way of studying how players arrive at an equilibrium point [Fudenberg and Levine, 1998]. As such, learning dynamics is prominently featured in a wide range of areas from evolutionary game theory [Weibull, 1997], to wireless network [Lasaulce and Tembine, 2011], to our topic of congestion games [Shah and Shin, 2010]. In general, learning algorithms may not converge, which leads us to two threads.

First, we consider linear and exponential cost functions with convergence guarantees [Klimm and Schütz, 2022]. We derive explicit running times for $k$-DCGs and their variants for these cost functions. Second, we consider the general

(potentially non-monotonic) cost functions with no convergence guarantees. We present approximation algorithms for this general case.

### LINEAR COST FUNCTIONS

We study an iterative best-response algorithm, where players start with an arbitrary strategy profile and iteratively play best responses until convergence to a PSNE, for $k$-DCGs and their variants using potential functions. Let the linear cost function of any resource $r$ under a strategy profile $\mathbf{s}$ be
$c_r(\mathbf{x}_r(\mathbf{s})) \equiv a_r \sum_{j \in [k]} z_j \mathbf{x}_{r,j}(\mathbf{s}) + b_r = a_r[\mathbf{z} \cdot \mathbf{x}_r(\mathbf{s})] + b_r$,
where $a_r, b_r \geq 0\ \forall r$ and the $k$-dimensional vector $\mathbf{z} \geq 0$.

We have the following results on $k$-DCGs and their variants. Notably, Klimm and Schütz [2022] provide an alternative proof of the existence of a potential function for this class of congestion games via the isomorphism technique. However, their proof is focused on existence and leaves open computational questions, especially for variants of $k$-DCGs, which we address here. The complete proofs are in the Appendix.

**Theorem 10.** *For linear-cost $k$-DCGs, the best-response algorithm runs in polynomial time if $\max_r a_r, \max_r b_r$, and $\frac{\max_i [\mathbf{z} \cdot \mathbf{d}_i]^2}{\min_i [\mathbf{z} \cdot \mathbf{d}_i]}$ are polynomial in $n$.*

**Theorem 11.** *For linear-cost $k$-DCGs with binary demand vectors vectors, the best-response algorithm runs in polynomial time if the following cost function parameters are polynomial in $n$: $\max_r a_r$, $\max_r b_r$, and $\max_j z_j$.*

**Theorem 12.** *For linear-cost $k$-CCGs, the best-response algorithm runs in polynomial time if $\max_r a_r$, $\max_r b_r$, $\frac{\max_j z_j^2}{\min_j z_j}$, and $\frac{\max_i d_{i,l(i)}^2}{\min_i d_{i,l(i)}}$ are polynomial in $n$, where $l(i) \in [k]$ denotes the index of the non-zero element in $\mathbf{d}_i$.*

Please note that polynomial-time algorithms for linear cost

are unlikely to exist due to the PLS-completeness of un-weighted network congestion games with linear cost [Ackermann et al., 2008].

### Experiments

We have performed experiments to evaluate the effect of the dimension $k$ and the number of resources on the running time of the algorithm given by Theorem 10. We vary $k = 2, 3, 4$ and the number of links $m$ in a parallel network from 2 to 10. We also vary the number of players $n$ from 5 to 100. In the iterative best-response algorithm, we apply a tweak suggested by Panagopoulou and Spirakis [2007] that prioritizes players with relatively high impacts on the cost function due to their demand vectors.

Our experiments show that the PSNE computation time of Theorem 10 scales up gracefully as we increase the number of players and links. This is perhaps not surprising given the pseudopolynomial running time of the algorithm. Furthermore, our experiments are consistent with those on single-dimensional weighted congestion games [Panagopoulou and Spirakis, 2007]. Details, including figures, are in the Appendix.

### EXPONENTIAL COST FUNCTIONS

For one-dimensional weighted congestion games, it has been shown that the uniform exponential cost function $c_r(x_r(\mathbf{s})) = \exp(x_r(\mathbf{s}))$ leads to a potential function [Panagopoulou and Spirakis, 2007]. For (one-dimensional) weighted congestion games, this result has been extended to non-uniform exponential functions of the shape $c_r(x_r(\mathbf{s})) = a_r \exp(x_r(\mathbf{s})) + b_r$ [Harks et al., 2011, Harks and Klimm, 2012]. For $k$-DCGs, it has been shown that games with cost functions of the shape $c_r(\mathbf{x}_r(\mathbf{s})) = a_r \exp(\mathbf{z} \cdot \mathbf{x}_r(\mathbf{s})) + b_r$ are isomorphic to one-dimensional congestion games [Klimm and Schütz, 2022]. For this cost function, we use Harks et al. [2011]'s results on 1-DCGs to derive a potential function for $k$-DCGs, which ultimately leads to the following result. Details, including the intermediate steps, are in the Appendix.

**Theorem 13.** *The best-response algorithm runs in polynomial time for exponential-cost $k$-DCGs if $\max_r a_r$ and $\max_r b_r$ are polynomial in $n$ and $[\mathbf{z} \cdot \mathbf{d}_N]$ is $\mathcal{O}(\log n)$.*

Since the cost function is exponential and an exponential term appears directly in the potential function, it is not surprising that in the above result, we need $[\mathbf{z} \cdot \mathbf{d}_N]$ to be $\mathcal{O}(\log n)$ for polynomial running time.

### APPROXIMATE PSNE FOR GENERAL COST FUNCTIONS

Very recently, several algorithms to compute approximate PSNE (in the multiplicative sense) have appeared. For $\alpha \geq 1$, an $\alpha$-PSNE $\mathbf{s}^*$ means that for any player $i$, $\pi_i(\mathbf{s}^*) \leq \alpha \pi_i(s_i', \mathbf{s}_{-i})$ for all $s_i'$. For polynomial cost functions of maximum degree $\delta$, an algorithm for computing a $(\delta + 1)$-approximate PSNE has been given in [Caragiannis and Fanelli, 2021]. This result has been extended to an $n$-PSNE algorithm for monotonic costs [Christodoulou et al., 2023]. The idea is to relate the decrease in cost due to any player's unilateral deviation to the decrease in social cost and reach a local minimum of the social cost.

We present an $(\alpha, \beta)$-PSNE algorithm for general cost. Let $\Delta_r \equiv \max\{\max_{i \in N, \mathbf{s} \in S; r \in s_i} c_r(\mathbf{x}_r(\mathbf{s}) - \mathbf{d}_i) - c_r(\mathbf{x}_r(\mathbf{s})), 0\}$ be the maximum non-negative marginal decrease of any player for resource $r$. When the congestion function is nondecreasing, $\Delta_r = 0$. Otherwise, $\Delta_r > 0$. Let $\Delta_{\max} = \max_{r \in R} \Delta_r$. The following result generalizes the result in Christodoulou et al. [2023] by removing the monotonicity assumption on the cost function while retaining the non-negative cost assumption.

**Theorem 14.** *Every $k$-DCG has an $(\alpha, \beta)$-PSNE for $\alpha = n$ and $\beta = (n-1)m\Delta_{\max}$. Furthermore, it can be computed using an iterative algorithm that is guaranteed to converge.*

In the iterative algorithm of Theorem 14, at each round, if $\pi_i(s_i, \mathbf{s}_{-i}) > n\pi_i(s_i', \mathbf{s}_{-i}) + (n-1)m\Delta_{\max}$ for any player $i$ currently playing $s_i$, $i$ deviates to $s_i'$. As the set of strategy profiles is finite, we eventually reach an $(\alpha, \beta)$-PSNE. The result is especially useful for small $\Delta_{\max}$ (e.g., noise).

## 6 STRUCTURED COSTS AND DEMANDS

Our study of structured costs and demands is motivated by a variety of realistic examples of traffic congestion games, where resources represent roads. As an example of structured/ordered demands, vehicles can be ordered by their demand vectors representing width, length, weight, etc. (e.g., semis, pickup trucks, SUVs, sedans, and so on). A common example of a nondecreasing cost function is more vehicles on the road means higher costs for everyone. Singleton strategies are seen in grid-patterned road networks with parallel roads to go from source to destination [Milchtaich, 2006]. We also consider structured cost functions– e.g., different types of roads have different speed limits: highways, county routes, local roads, etc.

### ORDERED DEMAND, NONDECREASING COST, AND SINGLETON STRATEGIES

Suppose that the players can be ordered according to their demand vectors: $\mathbf{d}_1 \geq \mathbf{d}_2 \geq ... \geq \mathbf{d}_n$ (w.l.o.g.). Let each

player $i$'s set of *singleton* strategies $S_i = \{\{r\} \mid r \in R\}$. In addition, assume that the cost functions are nondecreasing. We can compute a PSNE using the greedy best response algorithm, which orders the players from high to low demand and lets them play their best response in that order [Milchtaich, 2006]. Details are in the Appendix.

**Theorem 15.** *For a $k$-DCG with ordered demand vectors, nondecreasing cost functions, and singleton-resource strategies, a PSNE can be computed in $\mathcal{O}(n \log n + nmk)$ time.*

## ORDERED DEMAND, NONDECREASING COST, AND SHARED STRATEGIES

We relax the assumption of singleton-resource strategies. We show that as long as the players have the same set of strategies, we can compute a PSNE efficiently using the greedy best response algorithm.

**Theorem 16.** *For a $k$-DCG with ordered demand vectors, nondecreasing cost functions, and a shared set of strategies of size $p$, a PSNE can be computed in $\mathcal{O}(n \log n + npmk)$.*

## STRUCTURED COST FUNCTIONS AND SINGLETON STRATEGIES

In this scenario, we do not assume any ordering among the demands of the players. Instead, we assume that the cost functions are nondecreasing and that the resources are ordered by their cost functions. That is, w.l.o.g., $c_1(\mathbf{x}) \geq c_2(\mathbf{x}) \geq ... \geq c_m(\mathbf{x})$ for any $\mathbf{x}$. We also assume that there are constants $\alpha_j \geq 1$ such that $c_{j-1}(\mathbf{x}) = \alpha_j c_j(\mathbf{x})$ for any resource $j > 1$ and $\mathbf{x}$. These assumptions mean that some resources are more costly than others and that the costs of the resources are "nicely separated." Finally, we assume singleton-resource strategies. We get the following result.

**Theorem 17.** *For a $k$-DCG with nondecreasing and structured cost functions, where there are constants $\alpha_j \geq 1$ such that $c_{j-1}(\mathbf{x}) = \alpha_j c_j(\mathbf{x})$ for any resource $j > 1$ and aggregate demand vector $\mathbf{x}$, and singleton-resource strategies, a PSNE can be computed in $\mathcal{O}(n \log n + nmk)$ time.*

## 7 CONCLUSION

We have conducted a thorough computational study of $k$-DCGs and their variants using two different computational methods: CSP and learning dynamics. These two computational approaches are driven by whether or not a PSNE is guaranteed to exist in a class of $k$-DCGs. We prove the hardness of some very special cases and give polynomial-time algorithms for various problems under certain assumptions. Our CSP-based framework is applicable to general (potentially non-monotonic) cost functions for $k$-DCGs and their

variants. We also give pseudo-polynomial time algorithms based on learning dynamics for linear and exponential cost functions. We extend the learning dynamics approach to the study of approximation algorithms for general cost functions and exact algorithms for various types of structured demands and costs.

In particular, our CSP framework, which has not been studied before within the extremely rich congestion games literature, holds promise for future research within and outside of congestion games. We are particularly interested in designing and implementing CSP-inspired search algorithms for network congestion games, such as backjumping (Gaschnig, graph-based, conflict directed, etc.) and learning algorithms [Dechter, 2003], backtracking with tree decomposition [Jégou and Terrioux, 2003], AND/OR search algorithms [Marinescu and Dechter, 2009], etc. We are also interested in exploring some of the widely used solvers because to our knowledge, very large-scale experimental work is yet to be done on congestion games. Beyond the realm of congestion games, our key insight of decoupling players' strategies may have applications in many other game-theoretic problems.

## Acknowledgements

We thank the reviewers for their kind words and many helpful suggestions. MTI is grateful to the National Science Foundation for support from Award IIS-1910203. HC is supported by the National Institute of General Medical Sciences of the National Institutes of Health (P20GM130461), the Rural Drug Addiction Research Center at the University of Nebraska-Lincoln, and the National Science Foundation under grant IIS-2302999. The content is solely the responsibility of the authors and does not necessarily represent the official views of the funding agencies.

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

# Equilibrium Computation in Multidimensional Congestion Games: CSP and Learning Dynamics Approaches
# (Supplementary Material)

**Mohammad T. Irfan**[1]        **Hau Chan**[2]        **Jared Soundy**[2]

[1]Department of Computer Science, Bowdoin College, Brunswick, Maine, USA
[2]School of Computing, University of Nebraska-Lincoln, Lincoln, Nebraska, USA

Due to space constraints, we could only provide proof sketches for most of our results within the main text. We provide complete proofs here. We also provide additional details on our experiments. We minimally repeat some parts from the main text so that the reader can follow the proofs without having to switch back and forth between the main text.

The section numbers in this Appendix have been changed to the corresponding letter in the alphabetic ordering, but the theorem numbers are unchanged from the main text.

## C  COMPUTATIONAL COMPLEXITY

**Theorem 3.** *Deciding the existence of a PSNE in a $k$-DCG is NP-complete even when the demand vector $\mathbf{d}_i$ of each player $i \in N$ is a binary vector and $k$ is sublinear in the number of players. That is, $\mathbf{d}_i \in \{0,1\}^k$ for all $i$ and $k = \mathcal{O}(\log n)$.*

*Proof.* We first show that the problem is in NP. In particular, given any strategy profile $\mathbf{s} \in S$, we show that it can be verified in polynomial time that the profile is a PSNE. Observe that according to Definition 1, it is sufficient to check the potential deviation of each player $i \in N$. Since each player $i \in N$ has $|S_i|$ strategies and there are $n$ players, the verification takes at most some polynomial of the representation of the game.[1]

Next, we show that the considered problem is NP-hard by reducing from the problem of determining a PSNE in a weighted congestion game, which is known to be strongly NP-hard even for weighted network congestion games [Dunkel and Schulz, 2008].

More specifically, given a 1-DCG $\left(\check{n}, \widetilde{R}, \{\widetilde{S}_i, \widetilde{d}_i\}_{i \in \check{n}}, \{\widetilde{c}_r\}_{r \in \widetilde{R}}\right)$ with integer weights/demands bounded by some polynomial in the number of players (i.e., via Strongly NP-hardness), we construct a $k$-DCG $(N, R, \{S_i, \mathbf{d}_i\}_{i \in N}, \{c_r\}_{r \in R}, k)$ via the following:

- Let $N = \check{n}$ be the same set of $n$ players;
- Let $R = \widetilde{R}$ be the same set of $m$ resources;
- For each player $i \in N$, let $S_i = \widetilde{S}_i$ be the same set of strategies;
- Let $k = \lfloor \log \max_{i \in N} \widetilde{d}_i \rfloor + 1$ be the maximum length of the binary representation;
- For each player $i \in N$, let $\mathbf{d}_i = (d_{i_k}, ..., d_{i_1})$ be the binary demand vector induced by the binary representation of $\widetilde{d}_i$;
- For each resource $r \in R$ and $\mathbf{x} = (x_k, ..., x_1) \in \{0, 1, ..., n\}^k$, we define $c_r(\mathbf{x}) = \widetilde{c}_r\left(\sum_{j=1}^{k} 2^{(j-1)} x_j\right)$.

---

[1]Note that there are special types of 1-DCG (e.g., network congestion games) with more compact representations [Dunkel and Schulz, 2008]. The verification question for these types of games is in NP, and our hardness proof still holds for them.

Then for all $r \in R$, $\mathbf{s} = (s_1, ..., s_n) \in S$, and $\mathbf{x}_r(\mathbf{s}) = \sum_{i \in N; r \in s_i} \mathbf{d}_i$,

$$c_r(\mathbf{x}_r(\mathbf{s})) = c_r \left( \sum_{i \in N; r \in s_i} \mathbf{d}_i \right)$$

$$= c_r \left( \sum_{i \in N; r \in s_i} (d_{i_k}, ..., d_{i_1}) \right)$$

$$= \widetilde{c}_r \left( \sum_{j=1}^{k} 2^{(j-1)} \sum_{i \in N; r \in s_i} d_{i_j} \right)$$

$$= \widetilde{c}_r \left( \sum_{i \in N; r \in s_i} \sum_{j=1}^{k} 2^{(j-1)} d_{i_j} \right)$$

$$= \widetilde{c}_r \left( \sum_{i \in N; r \in s_i} \widetilde{d}_i \right) = \widetilde{c}_r \left( \widetilde{x}_r(\mathbf{s}) \right),$$

where $\widetilde{x}_r(\mathbf{s}) = \sum_{i \in N; r \in s_i} \widetilde{d}_i$, the first equality is by the definition of $\mathbf{x}_r(\mathbf{s})$, the second equality is by the definition of $\mathbf{d}_i$, the third equality is by our construction of the cost function, the fourth equality is by moving the sum over dimensions inside, and the fifth equality is because $\mathbf{d}_i$ is the binary representation of $\widetilde{d}_i$ by our construction.

Because of the above equivalence and the fact that $\pi_i(\mathbf{s}) = \sum_{r \in s_i} c_r(\mathbf{x}_r(\mathbf{s})) = \sum_{r \in s_i} \widetilde{c}_r(\widetilde{x}_r(\mathbf{s})) = \widetilde{\pi}_i(\mathbf{s})$ for all $r \in R$ and $\mathbf{s} \in S$, there is a PSNE in the 1-DCG instance if and only if it is a PSNE in the $k$-DCG instance.

We finally note that the reduction can be done in polynomial time. First, because the reduced 1-DCG instance is strongly NP-hard, the demands are polynomially bounded by $n$. Hence, $k = \mathcal{O}(\log n)$ is sublinear in the number of players by constructions. Second, converting an integer $x$ to its binary representation can be done in $\mathcal{O}(\log x)$ by repeatedly dividing $x$ and storing its remainders. Thus, constructing $\mathbf{d}_i$'s can be done in polynomial time. $\qquad\square$

**Theorem 4.** *Deciding the existence of a PSNE in a $k$-DCG (or a $k$-CCG) is NP-complete even when the demand vector $\mathbf{d}_i$ of each player $i \in N$ is a binary unit vector and $k$ is linear of the number of players. That is, $\mathbf{d}_i \in \{\mathbf{x} \in \{0,1\}^k ; \sum_{j=1}^{k} x_j = 1\}$ for all $i$ and $k = \mathcal{O}(n)$.*

*Proof.* The problem is clearly in NP because we can verify a PSNE in polynomial time.

To show the problem is NP-hard, we reduce from the problem of determining a PSNE in a weighted congestion game, which is known to be strongly NP-hard even for weighted network congestion games [Dunkel and Schulz, 2008].

More specifically, given a 1-DCG $(\overline{N}, \overline{R}, \{\overline{S}_i, \overline{d}_i\}_{i \in \overline{N}}, \{\overline{c}_r\}_{r \in \overline{R}})$ with integer weights/demands bounded by some polynomial in the number of players (i.e., via Strongly NP-hardness), we construct a $k$-DCG $(N, R, \{S_i, \mathbf{d}_i\}_{i \in N}, \{c_r\}_{r \in R}, k)$ via the following:

- Let $N = \overline{N}$ be the same set of $n$ players;
- Let $R = \overline{R}$ be the same set of $m$ resources;
- For each player $i \in N$, let $S_i = \overline{S}_i$ be the same set of strategies;
- Let $k = n$ be the number of dimensions corresponding to the number of players;
- For each player $i \in N$, we let $\mathbf{d}_i$ to be the binary unit demand vector of player $i$ of length $k$ where all entries are zero except the $i^{th}$ entry;
- For each resource $r \in R$ and $\mathbf{x} = (x_1, ..., x_k) \in \{0,1\}^k$, we define

$$c_r(\mathbf{x}) = \overline{c}_r \left( \sum_{j=1}^{k} \overline{d}_j x_j \right).$$

Following from the above construction, for all $r \in R$, strategy profile $\mathbf{s} = (s_1, ..., s_n) \in S$, and $\mathbf{x}_r(\mathbf{s}) = \sum_{i \in N; r \in s_i} \mathbf{d}_i$, we have

$$c_r(\mathbf{x}_r(\mathbf{s})) = c_r \left( \sum_{i \in N; r \in s_i} \mathbf{d}_i \right)$$

$$= c_r \left( \sum_{i \in N; r \in s_i} (d_{i_k}, ..., d_{i_1}) \right)$$

$$= \overline{c}_r \left( \sum_{j=1}^{k} \overline{d}_j \sum_{i \in N; r \in s_i} d_{i_j} \right)$$

$$= \overline{c}_r \left( \sum_{j=1}^{k} \overline{d}_j \mathbb{1}[r \in s_j] \right)$$

$$= \overline{c}_r \left( \sum_{i \in N; r \in s_i} \overline{d}_i \right)$$

$$= \overline{c}_r \left( \overline{x}_r(\mathbf{s}) \right),$$

where $\overline{x}_r(\mathbf{s}) = \sum_{i \in N; r \in s_i} \overline{d}_i$, $\mathbb{1}[r \in s_j]$ is an indicator function that returns 1 if the condition is true or 0 otherwise, the first equality is by the definition of $\mathbf{x}_r(\mathbf{s})$, the second equality is by the definition of $\mathbf{d}_i$, the third equality is by our construction of the cost function, the fourth equality is by noting that the demand of each player is zero for the $j^{th}$ entry except player $j$ for each dimension $j$ in our construction, and the fifth equality is because we account for players that use $r$.

Because of the above equivalence and the fact that $\pi_i(\mathbf{s}) = \sum_{r \in s_i} c_r(\mathbf{x}_r(\mathbf{s})) = \sum_{r \in s_i} \overline{c}_r(\overline{x}_r(\mathbf{s})) = \overline{\pi}_i(\mathbf{s})$ for all $r \in R$ and $\mathbf{s} \in S$, it is not hard to verify that there is a PSNE for one game if and only if it is a PSNE for the other. $\qquad\square$

## D    GENERAL COST: A CSP APPROACH

For the ease of reading, we repeat the description of Procedures 1 and 2 of the CSP approach from the main text.

**Procedure 1: Compute Domains of Dual Variables $v_{i,Y}$**

Given a configuration $\mathbf{y} \equiv (\mathbf{y}_1, \mathbf{y}_2, ..., \mathbf{y}_m)$, where each $\mathbf{y}_j \in \{0, ..., w_{\max}\}^k$, we compute the set of strategies for each player $i$ that makes $i$ "happy" under the configuration. To do this, abusing the notation $\pi_i$ slightly, we define and compute, for any $i \in N$, $s_i, s_i' \in S_i$, and $s_i \neq s_i'$,

$$\pi_i(s_i, \mathbf{y}) = \sum_{r \in s_i} c_r(\mathbf{y}_r)$$

$$\pi_i(s_i, \mathbf{y}, s_i') = \sum_{r \in s_i' \cap s_i} c_r(\mathbf{y}_r) + \sum_{r \in s_i' \setminus s_i} c_r(\mathbf{y}_r + \mathbf{d}_i)$$

$$BR_i(\mathbf{y}) = \{s_i \in S_i \mid \forall s_i' \in S_i, \pi_i(s_i, \mathbf{y}) \leq \pi_i(s_i, \mathbf{y}, s_i')\}$$

The first equation calculates player $i$'s cost. The second calculates player $i$'s cost when deviating from $s_i$ (under $\mathbf{y}$) to $s_i'$. $BR_i(\mathbf{y})$ computed in the last equation is the set of $i$'s best responses to $\mathbf{y}$. Therefore, the domain of $v_{i,Y}$ is the union of sets $\{(s_i, \mathbf{y}) \mid s_i \in BR_i(\mathbf{y})\}$ for all $\mathbf{y}$.

We deliberately do not compute the domain of $v_{N,Y}$ (the dual variable for the primal feasibility constraint) because it may contain numerous strategy profiles that are not PSNE. We next show in Procedure 2 how we can search for a PSNE without explicitly computing the domain of $v_{N,Y}$.

**Procedure 2: Search for PSNE**

Given a configuration $\mathbf{y} \equiv (\mathbf{y}_1, \mathbf{y}_2, ..., \mathbf{y}_m)$, a PSNE under it is a strategy profile $\mathbf{s} = (s_1, ..., s_n)$ such that (1) $(s_i, \mathbf{y})$ is in the domain of $v_{i,Y}$ for each player $i$, and (2) $\mathbf{x}_1(\mathbf{s}) = \mathbf{y}_1, \mathbf{x}_2(\mathbf{s}) = \mathbf{y}_2, ..., \mathbf{x}_m(\mathbf{s}) = \mathbf{y}_m$. The first condition enforces players' best responses to $\mathbf{y}$, while the second condition enforces the feasibility constraint. We get the following general result.

**Theorem 5.** *For any $k$-DCG, there is an algorithm to determine the existence of a PSNE in $\mathcal{O}((w_{\max})^{km}(nkp^2m^2 + nkmp(w_{\max})^{km}))$. The algorithm is polynomial in $n$, $p$, and $w_{\max}$, when $m$ and $k$ are constants.*

*Proof.* For each configuration $\mathbf{y}_1, \mathbf{y}_2, ..., \mathbf{y}_m \in \{0, ..., w_{\max}\}^k$, we perform Procedures 1 and 2 to verify and construct if there is any PSNE that is consistent with the configuration.

Regarding Procedure 1, for each player $i \in N$, computing $BR_i$ takes at most $\mathcal{O}(kp^2m^2)$ time for each configuration. This is because the first equation takes $\mathcal{O}(m)$ time for a given configuration and $s_i$, and the second equation takes $\mathcal{O}(km^2)$ for a given configuration, $s_i$, and $\overline{s}_i$. Thus, this procedure's overall running time is $\mathcal{O}(nkp^2m^2)$ for all players.

Procedure 2 can be done efficiently using dynamic programming, where we (1) first order the players $1, ..., n$ and (2) create a binary table $T_i(\mathbf{y}'_1, \mathbf{y}'_2, ..., \mathbf{y}'_m) \in \{0, 1\}$ for each $\mathbf{y}'_1, \mathbf{y}'_2, ..., \mathbf{y}'_m \in \{0, ..., w_{\max}\}^k$ of size $\mathcal{O}((w_{\max})^{km})$ for each player $i$. We first initialize $T_0(\mathbf{0}, ..., \mathbf{0}) = 1$ where we have an all zero configuration. We then define $T_i(\mathbf{y}'_1, \mathbf{y}'_2, ..., \mathbf{y}'_m) = 1$ if and only if there is $\overline{\mathbf{y}}_1, \overline{\mathbf{y}}_2, ..., \overline{\mathbf{y}}_m$ such that $T_{i-1}(\overline{\mathbf{y}}_1, \overline{\mathbf{y}}_2, ..., \overline{\mathbf{y}}_m) = 1$ and, for some $s_i \in BR_i(\mathbf{y}_1, \mathbf{y}_2, ..., \mathbf{y}_m)$, $\mathbf{y}'_r = \overline{\mathbf{y}}_r + \mathbb{1}[r \in s_i]\mathbf{d}_i$ for each $r \in R$. Table $T_i$ can be constructed by looking at all the 1's entries of $T_{i-1}$ and adding the player demand vector to the corresponding resources for each $s_i \in BR_i(\mathbf{y}_1, \mathbf{y}_2, ..., \mathbf{y}_m)$.

Because there are at most $\mathcal{O}((w_{\max})^{km})$ configurations, and each of the $n$ players has at most $p$ strategies with size at most $m$ and at most $k$ dimensions, the time for this procedure is at most $\mathcal{O}(nkmp(w_{\max})^{km})$. To verify whether a given $\mathbf{y}_1, \mathbf{y}_2, ..., \mathbf{y}_m$ can be achieved, one can check if $T_n(\mathbf{y}_1, \mathbf{y}_2, ..., \mathbf{y}_m)$ is 1, in which case a corresponding PSNE can be constructed via the standard tracing back procedure of dynamic programming.

The total time (Procedures 1 and 2) to check a given configuration can be formed as a PSNE is $\mathcal{O}(nkp^2m^2 + nkmp(w_{\max})^{km})$. Thus, to verify all configurations, the total time is $\mathcal{O}((w_{\max})^{km}(nkp^2m^2 + nkmp(w_{\max})^{km}))$, which is polynomial in $n$, $p$, and $w_{\max}$, when $m$ and $k$ are constants.

If there is such a strategy profile for some configuration, then the game has a PSNE. Otherwise, the game does not have any PSNE. The reason is that each PSNE must correspond to some configuration, and we enumerate each configuration to search for a PSNE. For a given configuration that corresponds to a PSNE, each player $i$'s equilibrium strategy must be in $BR_i$ because $BR_i$ contains all strategies in which player $i$ does not have any incentive to deviate to other strategies from the configuration. We note that there are some configurations that might not be feasible (e.g., some $\mathbf{y}_r$ that are too small or $\mathbf{y}_r + \mathbf{d}_i$ outside of the $w_{\max}$). The above procedure would eliminate them when searching for a PSNE, thereby removing configurations that are not consistent with any strategy profiles. $\square$

## VARIANT: BINARY DEMAND VECTORS

In Section 3, we showed that $k$-DCGs with binary demand vectors are provably hard. We can still apply Theorem 5 to derive a pseudopolynomial time algorithm when $k$ and $m$ are bounded. However, an improved analysis gives us the following result. Note that in the case of binary demand vectors, any $j$-th element of an aggregated demand vector corresponds to the number of players having the $j$-th bit of their demand vector "on." Therefore, for clarity, we use $\check{n} = \max_{j \in [k]} \sum_{i \in N} d_{i_j}$ in place of $w_{\max}$ to denote the maximum number of players having a demand vector bit on. The following result is particularly interesting when $\check{n} \ll n$.

**Theorem 6.** *For $k$-DCGs with binary demand, there is an $\mathcal{O}(\check{n}^{km}(nkp^2m^2 + \min\{nkmp\check{n}^{km}, n^{km+1}p\}))$-time algorithm to compute a PSNE or decide none exists. The algorithm is polynomial in $n$ and $p$ when $m$ and $k$ are constants.*

*Proof.* Due to the binary demand vectors, an element of the $k$-dimensional aggregated demand vector can be at most $n$. Putting $w_{\max} = \widetilde{n}$, as a corollary of Theorem 5, the running time of the algorithm is $\mathcal{O}(\widetilde{n}^{km}(nkp^2m^2 + nkmp\widetilde{n}^{km}))$. However, using the structure of the game, we can shave off a factor of $km$ from the second term, as shown below.

Here, we focus on the running time of Procedure 2. Recall that we are given a configuration $\mathbf{y}_1, \mathbf{y}_2, ..., \mathbf{y}_m$ and $BR_i(\mathbf{y}_1, \mathbf{y}_2, ..., \mathbf{y}_m)$ for each player $i$. We want to pick a strategy from $BR_i$ for each player $i$ so that the aggregated demand of the picked strategy profile is exactly $\mathbf{y}_1, \mathbf{y}_2, ..., \mathbf{y}_m$. As usual, we start with an all zero configuration and define

$T_0(\mathbf{0}, ..., \mathbf{0}) = 1$. We then go over the players $1, ..., n$, one at a time. Observe that when we consider player 1, the number of configurations (or table entries) for $T_1$ is at most $2^{km}$ because the elements of the $k$-dimensional vector for each of the $m$ resources are either 0 or 1. In this fashion, when we consider player $i$, the number of configurations for $T_i$ is at most $(i+1)^{km}$. We initialize $T_i$ to 0 for these configurations in $\mathcal{O}((i+1)^{km})$ time. We set $T_i(\mathbf{y}'_1, \mathbf{y}'_2, ..., \mathbf{y}'_m) = 1$ if and only if there is $\overline{\mathbf{y}}_1, \overline{\mathbf{y}}_2, ..., \overline{\mathbf{y}}_m$ such that $T_{i-1}(\overline{\mathbf{y}}_1, \overline{\mathbf{y}}_2, ..., \overline{\mathbf{y}}_m) = 1$ and, for some $s_i \in BR_i(\mathbf{y}_1, \mathbf{y}_2, ..., \mathbf{y}_m)$, $\mathbf{y}'_r = \overline{\mathbf{y}}_r + \mathbb{1}[r \in s_i]\mathbf{d}_i$ for each $r$. Note that there are $i^{km}$ possibilities of $\overline{\mathbf{y}}_1, \overline{\mathbf{y}}_2, ..., \overline{\mathbf{y}}_m$ from player $i-1$.

Therefore, the running time of Procedure 2 is $\mathcal{O}\left(\sum_{i=1}^n \left[(i+1)^{km} + kpmi^{km}\right]\right)$, which is dominated by $\mathcal{O}\left(\sum_{i=1}^n kpmi^{km}\right) = \mathcal{O}\left(kpm\sum_{i=1}^n i^{km}\right)$. Here, $\sum_{i=1}^n i^{km}$ is $\mathcal{O}\left(\frac{n^{km+1}}{km+1}\right)$. As a result, the running time of Procedure 2 is $\mathcal{O}(n^{km+1}p)$.

The running time of Procedure 1 is unchanged from Theorem 5. Since both Procedures 1 and 2 are run for each of the $\widetilde{n}^{km}$ configurations, the total running time is $\mathcal{O}(\widetilde{n}^{km}(nkp^2m^2 + \min\{nkmp\widetilde{n}^{km}, n^{km+1}p\}))$. $\quad\square$

## VARIANT: $k$-CLASS CONGESTION GAME ($k$-CCG)

Let the *class of player $i$* be the index where the positive element appears in $\mathbf{d}_i$. Although Theorem 5 can be directly applied to this case, we can exploit the structure of the game to improve the running time. The key intuition is that the players can be partitioned according to their classes. The players in a class $j \in [k]$ can only affect the $j$-th index of the aggregated demand on any resource. That is, they affect the $j$-th index of each of $\mathbf{y}_1, \mathbf{y}_2, ..., \mathbf{y}_m$. As a result, Procedure 2 can be broken into $k$ different computational tasks, each corresponding to a class. This idea leads us to the following result. *Notably, compared to Theorem 5, this partition-based algorithm removes a $k$ term from the exponent.*

**Theorem 7.** *For $k$-CCGs, there is an $\mathcal{O}((w_{\max})^{km}(np^2m^2 + nkpm(w_{\max})^m))$ algorithm to compute a PSNE or decide none exists. The algorithm is polynomial in $n$, $p$, and $w_{\max}$ when $m$ and $k$ are constants.*

*Proof.* When we apply Procedure 1 to $k$-CCG, computing $\pi_i(s_i, \mathbf{y})$, $\pi_i(s_i, \mathbf{y}, s'_i)$, and $BR_i$ (using the three equations in Procedure 1) for each player $i$ takes $\mathcal{O}(m)$, $\mathcal{O}(m^2)$, and $\mathcal{O}(p^2m^2)$, respectively. The saving of a factor of $k$ compared to Theorem 5 is due to the addition in the second equation for $\pi_i(s_i, \mathbf{y}, s'_i)$ being basically one-dimensional as opposed to $k$-dimensional. Thus, Procedure 1 runs in $\mathcal{O}(np^2m^2)$ time.

We now focus on Procedure 2. As a preprocessing step, we partition the players into $C_1, ..., C_k$ based on their classes. The players in $C_j$ can only affect the $j$-th element of the aggregated demand on any resource. Recall that in Procedure 2, we are given a configuration $\mathbf{y}_1, \mathbf{y}_2, ..., \mathbf{y}_m$. For each class $j \in [k]$, we construct a vector $\mathbf{y}^j = [y_{1j}, y_{2j}, ..., y_{mj}]$ to be used by the players in $C_j$. We now do the following operations in each partition $C_j$ independently.

We start the DP by ordering the players in $C_j$ as $1, 2, ..., |C_j|$ (wlog). We then create a binary table $T_i(z_1, z_2, ..., z_m) \in \{0, 1\}$ for each $z_1, z_2, ..., z_m \in \{0, ..., w_{\max}\}$ of size $\mathcal{O}((w_{\max})^m)$ for each player $i$ in $C_j$. We initialize $T_0(0, ..., 0) = 1$. We then define $T_i(z_1, z_2, ..., z_m) = 1$ if and only if there is $z'_1, z'_2, ..., z'_m$ such that $T_{i-1}(z'_1, z'_2, ..., z'_m) = 1$ and for some $s_i \in BR_i(\mathbf{y}_1, \mathbf{y}_2, ..., \mathbf{y}_m)$, $z_r = z'_r + \mathbb{1}[r \in s_i]d_{ij}$ for each $r \in R$.

Once we finish the table construction in all partitions, we have a PSNE if and only if for each partition $C_j$, $T_{|C_j|}(\mathbf{y}^j) = 1$. The argument is similar to the proof of Theorem 5, only that we have to collate $m$-dimensional vectors $\mathbf{y}^j$ for each class $j \in [k]$ to form the given configuration $\mathbf{y}_1, \mathbf{y}_2, ..., \mathbf{y}_m$.

Here, the running time of Procedure 2 is $\mathcal{O}(nkpm(w_{\max})^m)$. This is because in each of the $k$ partitions, we do $\mathcal{O}(npm(w_{\max})^m)$ work due to at most $n$ players in that partition, at most $p$ best-response strategies for each player, $m$ resources, and $(w_{\max})^m$ table entries.

Since both Procedures 1 and 2 are performed for each of the $(w_{\max})^{km}$ configurations, the total running time of the algorithm is $\mathcal{O}((w_{\max})^{km}(np^2m^2 + nkpm(w_{\max})^m))$. Note that the running time has an exponential saving of $k$ in the second term compared to Theorem 5. $\quad\square$

**Corollary 8.** *For $k$-CCGs with binary demand vectors, there is an $\mathcal{O}((\check{n})^{km}(np^2m^2 + nkpm(\check{n})^m))$-time algorithm to compute a PSNE or decide none exists. The algorithm is polynomial in $n$ and $p$ when $m$ and $k$ are constants.*

*Proof.* The argument is similar to the proof of Theorem 5, only that we have to collate $m$-dimensional vectors $\mathbf{y}^j$ for each class $j \in [k]$ to form the given configuration $\mathbf{y}_1, \mathbf{y}_2, ..., \mathbf{y}_m$.

Here, the running time of Procedure 2 is $\mathcal{O}(nkpm(w_{\max})^m)$. This is because in each of the $k$ partitions, we do $\mathcal{O}(npm(w_{\max})^m)$ work due to at most $n$ players in that partition, at most $p$ best-response strategies for each player, $m$ resources, and $(w_{\max})^m$ table entries.

Since both Procedures 1 and 2 are performed for each of the $(w_{\max})^{km}$ configurations, the total running time of the algorithm is $\mathcal{O}((w_{\max})^{km}(np^2m^2 + nkpm(w_{\max})^m))$. Note that the running time has an exponential saving of $k$ in the second term compared to Theorem 5. $\qquad\square$

## EXPERIMENTS

The algorithm given in theorem 5 is theoretically efficient under some assumptions. Here, we show that it is practically efficient. To our knowledge, brute force is the only other algorithm guaranteed to work on games of interest: multi-dimensional congestion games with non-monotonic cost functions. First, we compare two implementations of our algorithm against brute force. Our algorithm overtakes brute-force at a relatively small value of $n$. Second, we compare the implementations against the simulated worst-case complexity of the algorithm: $\mathcal{O}((w_{\max})^{km}(nkp^2m^2 + nkmp(w_{\max})^{km}))$. This shows that, in practice, our algorithm greatly outperforms its asymptotic behavior.

All algorithms were implemented in Python. Source code and data can be found in the supplementary material. Results were obtained on a Linux machine with an Intel® Xeon® E3-1225 @ 3.1 GHz and 24GB of RAM.

### Game Generation

The evaluation was done on a $k$-dimensional parallel link model with $m$ links or resources. Every player chooses one link $r \in R$ from the set of all $m$ links. Each link $r \in R$ had a non-monotonic cost function of $c_r(\mathbf{x}_r(\mathbf{s})) = \alpha_r f_r(\mathbf{x}_r(\mathbf{s})) + \beta_r$. Where $\alpha_r$ and $\beta_r$ are integers drawn uniformly randomly from the range $[0, 100]$. The non-monotonic component is $f_r(\mathbf{x}_r(\mathbf{s})) = f_r^1(\mathbf{x}_r^1(\mathbf{s})) + f_r^2(\mathbf{x}_r^2(\mathbf{s})) + \cdots + f_r^k(\mathbf{x}_r^k(\mathbf{s}))$, where $\mathbf{x}_r^j(\mathbf{s})$ is the aggregate demand in the $j$th dimension and $f_r^j$ is the cost of the aggregate demand in the $j$th dimension. The cost of $f_r^j$ for any given input is an integer drawn uniformly randomly from the range $[0, 100]$. Every element of the demand vector $d_{ij}$ was an integer drawn uniformly randomly from the range $[0, q]$. If every element of the demand vector was 0 then the entire demand vector was discarded and randomly generated again. For each combination of parameters $(m, k, q)$, 15 games were randomly generated and then $n$ players were randomly generated, all using the master seed 2024.

### Methods

The dynamic program was implemented in two ways. The first method is as described in section 4. The second method exploits the sparsity of 1's in the binary table, by replacing the binary table with a hashset. Both implementations contain the optimization where if a single player is found to have no best response for a configuration (Procedure 1, section 4) then the algorithm will stop computations on that configuration. Likewise the brute force implementation has the optimization where as soon as a single player is found who is willing to deviate from a strategy profile then computations for that strategy profile will stop. For each $n$ the time to enumerate all configurations or strategy profiles (respectively) was measured and averaged across each of the 15 games. The binary table dynamic program was also constrained by memory, so that if a level of the binary table consumed more than 1 GB of memory for a single game then execution for that parameter combination would be halted.

In order to chart the asymptotic behavior of the binary table dynamic program we had to ensure that it ran at its big-O speed not faster. First, all mentioned optimizations were removed. Second, because the asymptotic behavior of the algorithm in theorem 5 is based on the size of the binary table, all bits of the binary table were set to 1. To approximate the speed of the asymptotic binary table algorithm at a large $n$ the average time to check if a configuration contains a NE was measured separately for both procedure 1 $z_1$ and procedure 2 $z_2$. This was done because of memory and time constraints related to binary table size, which only affected procedure 2. The binary table size was forced to 1000 for each $n$. The average time was multiplied by $(nq)^{km}(z_1 + \frac{z_2(nq+1)^{km}}{1000})$ to approximate the asymptotic runtime.

# E  LEARNING DYNAMICS APPROACH

**Appendix Definition 1** (w-potential game [Monderer and Shapley, 1996])**.** *Given a vector of positive numbers* $\mathbf{w} = (w_i)_{i \in N} > 0$, *a game is called a* **w**-*potential game if it admits function* $P : S \to \mathbb{R}$ *such that for every* $i \in N$ *and for every* $\mathbf{s}_{-i}$, *the following holds for any* $s_i$ *and* $s_i'$.

$$\pi_i(s_i, s_{-i}) - \pi_i(s_i', s_{-i}) = w_i \cdot (P(s_i, s_{-i}) - P(s_i', s_{-i})).$$

*Here,* $P$ *is called a* **w**-*potential function.*

**LINEAR COST FUNCTIONS**

Let the linear cost function of any resource $r$ under a strategy profile $\mathbf{s}$ be $c_r(\mathbf{x}_r(\mathbf{s})) \equiv a_r \sum_{j \in [k]} z_j \mathbf{x}_{r,j}(\mathbf{s}) + b_r = a_r[\mathbf{z} \cdot \mathbf{x}_r(\mathbf{s})] + b_r$, where $a_r, b_r \geq 0 \ \forall r$ and the $k$-dimensional vector $\mathbf{z} \geq 0$.

Basically, resource $r$'s cost function is a weighted sum of the $k$ elements of the aggregated demand vector $\mathbf{x}_r(\mathbf{s})$ with resource-specific multiplicative and additive terms $a_r$ and $b_r$, respectively.

Let the linear cost function of any resource $r$ under a strategy profile $\mathbf{s}$ be $c_r(\mathbf{x}_r(\mathbf{s})) \equiv a_r \sum_{j \in [k]} z_j \mathbf{x}_{r,j}(\mathbf{s}) + b_r = a_r[\mathbf{z} \cdot \mathbf{x}_r(\mathbf{s})] + b_r$, where $a_r, b_r \geq 0$ for all $r$ and the $k$-dimensional vector $\mathbf{z} \geq 0$. Basically, resource $r$'s cost function is a weighted sum of the $k$ elements of the aggregated demand vector $\mathbf{x}_r(\mathbf{s})$ with resource-specific multiplicative and additive terms $a_r$ and $b_r$, respectively.

We study an iterative *best-response algorithm*, where players iteratively play best responses until convergence to a PSNE, for several variants of $k$-DCGs based on bounding a potential function. This algorithm starts with an arbitrary strategy profile. As long as some player can improve their cost, their best response is updated.

The next theorem presents a potential function for linear cost. Notably, Klimm and Schütz [2022] provide an alternative proof of this theorem via the isomorphism technique, but their proof is focused on existence and leaves open computational questions, which we address here.

**Appendix Theorem 2.** *Any multidimensional congestion game with linear resource costs is a* **w**-*potential game.*

*Proof.* The proof follows the same line of argument as the single dimensional case [Fotakis et al., 2005]. Here, the main task is to devise a potential function when the demands are vectors instead of scalars.

We show that the $\Phi(\mathbf{s})$ defined below is a **w**-potential function for the choice of $w_i = \frac{1}{2[\mathbf{z} \cdot \mathbf{d}_i]}$ for each $i$.

$$\Phi_1(\mathbf{s}) = \sum_{r \in R} c_r(\mathbf{x}_r(\mathbf{s}))[\mathbf{z} \cdot \mathbf{x}_r(\mathbf{s})].$$

$$\Phi_2(\mathbf{s}) = \sum_{i \in N} \sum_{r \in s_i} c_r(\mathbf{d}_i)[\mathbf{z} \cdot \mathbf{d}_i].$$

$$\Phi(\mathbf{s}) = \Phi_1(\mathbf{s}) + \Phi_2(\mathbf{s}). \tag{3}$$

Consider any set of resources $s_i' \neq s_i$. Define $\mathbf{s}' = (s_{-i}, s_i')$. For any resource $r$ that is picked either by both $s_i$ and $s_i'$ or none of them:

$$c_r(\mathbf{x}_r(\mathbf{s})) = c_r(\mathbf{x}_r(\mathbf{s}')). \tag{4}$$

$$c_r(\mathbf{x}_r(\mathbf{s}))\mathbf{x}_r(\mathbf{s}) = c_r(\mathbf{x}_r(\mathbf{s}'))\mathbf{x}_r(\mathbf{s}'). \tag{5}$$

For any $r \in s_i \setminus s_i'$:

$$
\begin{aligned}
&c_r(\mathbf{x}_r(\mathbf{s})) - c_r(\mathbf{x}_r(\mathbf{s}')) \\
&= a_r[\mathbf{z} \cdot \mathbf{x}_r(\mathbf{s})] + b_r - a_r[\mathbf{z} \cdot \mathbf{x}_r(\mathbf{s}')] - b_r \\
&= a_r[\mathbf{z} \cdot (\mathbf{x}_r(\mathbf{s}) - \mathbf{x}_r(\mathbf{s}'))] \\
&= a_r[\mathbf{z} \cdot \mathbf{d}_i], \text{ and}
\end{aligned}
$$

$$c_r(\mathbf{x}_r(\mathbf{s}))\mathbf{x}_r(\mathbf{s}) - c_r(\mathbf{x}_r(\mathbf{s}'))\mathbf{x}_r(\mathbf{s}')$$
$$= (a_r[\mathbf{z} \cdot \mathbf{x}_r(\mathbf{s})] + b_r)\mathbf{x}_r(\mathbf{s}) - (a_r[\mathbf{z} \cdot \mathbf{x}_r(\mathbf{s}')] + b_r)\mathbf{x}_r(\mathbf{s}')$$
$$= (a_r[\mathbf{z} \cdot \mathbf{x}_r(\mathbf{s})] + b_r)\mathbf{x}_r(\mathbf{s}) - (a_r[\mathbf{z} \cdot (\mathbf{x}_r(\mathbf{s}) - \mathbf{d}_i)] + b_r)(\mathbf{x}_r(\mathbf{s}) - \mathbf{d}_i)$$
$$= a_r[\mathbf{z} \cdot \mathbf{x}_r(\mathbf{s})]\mathbf{x}_r(\mathbf{s}) + b_r\mathbf{x}_r(\mathbf{s}) - a_r[\mathbf{z} \cdot \mathbf{x}_r(\mathbf{s})]\mathbf{x}_r(\mathbf{s}) + a_r[\mathbf{z} \cdot \mathbf{d}_i]\mathbf{x}_r(\mathbf{s}) - b_r\mathbf{x}_r(\mathbf{s})$$
$$\quad + a_r[\mathbf{z} \cdot \mathbf{x}_r(\mathbf{s})]\mathbf{d}_i - a_r[\mathbf{z} \cdot \mathbf{d}_i]\mathbf{d}_i + b_r\mathbf{d}_i.$$
$$= a_r[\mathbf{z} \cdot \mathbf{d}_i]\mathbf{x}_r(\mathbf{s}) + a_r[\mathbf{z} \cdot \mathbf{x}_r(\mathbf{s})]\mathbf{d}_i - a_r[\mathbf{z} \cdot \mathbf{d}_i]\mathbf{d}_i + b_r\mathbf{d}_i$$

Similarly, for any resource $r \in s_i' \setminus s_i$,

$$c_r(\mathbf{x}_r(\mathbf{s})) - c_r(\mathbf{x}_r(\mathbf{s}')) = -a_r[\mathbf{z} \cdot \mathbf{d}_i], \text{ and}$$

$$c_r(\mathbf{x}_r(\mathbf{s}))\mathbf{x}_r(\mathbf{s}) - c_r(\mathbf{x}_r(\mathbf{s}'))\mathbf{x}_r(\mathbf{s}')$$
$$= (a_r[\mathbf{z} \cdot \mathbf{x}_r(\mathbf{s})] + b_r)\mathbf{x}_r(\mathbf{s}) - (a_r[\mathbf{z} \cdot \mathbf{x}_r(\mathbf{s}')] + b_r)\mathbf{x}_r(\mathbf{s}')$$
$$= (a_r[\mathbf{z} \cdot \mathbf{x}_r(\mathbf{s})] + b_r)\mathbf{x}_r(\mathbf{s}) - (a_r[\mathbf{z} \cdot (\mathbf{x}_r(\mathbf{s}) + \mathbf{d}_i)] + b_r)(\mathbf{x}_r(\mathbf{s}) + \mathbf{d}_i)$$
$$= a_r[\mathbf{z} \cdot \mathbf{x}_r(\mathbf{s})]\mathbf{x}_r(\mathbf{s}) + b_r\mathbf{x}_r(\mathbf{s}) - a_r[\mathbf{z} \cdot \mathbf{x}_r(\mathbf{s})]\mathbf{x}_r(\mathbf{s}) - a_r[\mathbf{z} \cdot \mathbf{d}_i]\mathbf{x}_r(\mathbf{s}) - b_r\mathbf{x}_r(\mathbf{s})$$
$$\quad - a_r[\mathbf{z} \cdot \mathbf{x}_r(\mathbf{s})]\mathbf{d}_i - a_r[\mathbf{z} \cdot \mathbf{d}_i]\mathbf{d}_i - b_r\mathbf{d}_i$$
$$= -a_r[\mathbf{z} \cdot \mathbf{d}_i]\mathbf{x}_r(\mathbf{s}) - a_r[\mathbf{z} \cdot \mathbf{x}_r(\mathbf{s})]\mathbf{d}_i - a_r[\mathbf{z} \cdot \mathbf{d}_i]\mathbf{d}_i - b_r\mathbf{d}_i.$$

The difference in the $\Phi_1$ function under $\mathbf{s}$ and $\mathbf{s}'$ is

$$\Phi_1(\mathbf{s}) - \Phi_1(\mathbf{s}')$$
$$= \sum_{r \in R} \Big(c_r(\mathbf{x}_r(\mathbf{s}))[\mathbf{z} \cdot \mathbf{x}_r(\mathbf{s})] - c_r(\mathbf{x}_r(\mathbf{s}'))[\mathbf{z} \cdot \mathbf{x}_r(\mathbf{s}')]\Big)$$
$$= \mathbf{z} \cdot \sum_{r \in R} \Big(c_r(\mathbf{x}_r(\mathbf{s}))\mathbf{x}_r(\mathbf{s}) - c_r(\mathbf{x}_r(\mathbf{s}'))\mathbf{x}_r(\mathbf{s}')\Big)$$
$$= \mathbf{z} \cdot \sum_{r \in s_i \setminus s_i'} \Big(c_r(\mathbf{x}_r(\mathbf{s}))\mathbf{x}_r(\mathbf{s}) - c_r(\mathbf{x}_r(\mathbf{s}'))\mathbf{x}_r(\mathbf{s}')\Big) + \mathbf{z} \cdot \sum_{r \in s_i' \setminus s_i} \Big(c_r(\mathbf{x}_r(\mathbf{s}))\mathbf{x}_r(\mathbf{s}) - c_r(\mathbf{x}_r(\mathbf{s}'))\mathbf{x}_r(\mathbf{s}')\Big)$$
$$= \mathbf{z} \cdot \sum_{r \in s_i \setminus s_i'} \Big(a_r[\mathbf{z} \cdot \mathbf{d}_i]\mathbf{x}_r(\mathbf{s}) + a_r[\mathbf{z} \cdot \mathbf{x}_r(\mathbf{s})]\mathbf{d}_i - a_r[\mathbf{z} \cdot \mathbf{d}_i]\mathbf{d}_i + b_r\mathbf{d}_i\Big) -$$
$$\quad \mathbf{z} \cdot \sum_{r \in s_i' \setminus s_i} \Big(a_r[\mathbf{z} \cdot \mathbf{d}_i]\mathbf{x}_r(\mathbf{s}) + a_r[\mathbf{z} \cdot \mathbf{x}_r(\mathbf{s})]\mathbf{d}_i + a_r[\mathbf{z} \cdot \mathbf{d}_i]\mathbf{d}_i + b_r\mathbf{d}_i\Big)$$
$$= \sum_{r \in s_i \setminus s_i'} \Big(a_r[\mathbf{z} \cdot \mathbf{d}_i][\mathbf{z} \cdot \mathbf{x}_r(\mathbf{s})] + a_r[\mathbf{z} \cdot \mathbf{x}_r(\mathbf{s})][\mathbf{z} \cdot \mathbf{d}_i] - a_r[\mathbf{z} \cdot \mathbf{d}_i][\mathbf{z} \cdot \mathbf{d}_i] + b_r[\mathbf{z} \cdot \mathbf{d}_i]\Big) -$$
$$\quad \sum_{r \in s_i' \setminus s_i} \Big(a_r[\mathbf{z} \cdot \mathbf{d}_i][\mathbf{z} \cdot \mathbf{x}_r(\mathbf{s})] + a_r[\mathbf{z} \cdot \mathbf{x}_r(\mathbf{s})][\mathbf{z} \cdot \mathbf{d}_i] + a_r[\mathbf{z} \cdot \mathbf{d}_i][\mathbf{z} \cdot \mathbf{d}_i] + b_r[\mathbf{z} \cdot \mathbf{d}_i]\Big).$$
$$= \sum_{r \in s_i \setminus s_i'} \Big(2a_r[\mathbf{z} \cdot \mathbf{d}_i][\mathbf{z} \cdot \mathbf{x}_r(\mathbf{s})] - a_r[\mathbf{z} \cdot \mathbf{d}_i]^2 + b_r[\mathbf{z} \cdot \mathbf{d}_i]\Big) - \sum_{r \in s_i' \setminus s_i} \Big(2a_r[\mathbf{z} \cdot \mathbf{d}_i][\mathbf{z} \cdot \mathbf{x}_r(\mathbf{s})] + a_r[\mathbf{z} \cdot \mathbf{d}_i]^2 + b_r[\mathbf{z} \cdot \mathbf{d}_i]\Big).$$

The difference in the $\Phi_2$ function under $\mathbf{s}$ and $\mathbf{s}'$ is

$$
\Phi_2(\mathbf{s}) - \Phi_2(\mathbf{s}')
$$
$$
= \sum_{l \in N} \sum_{r \in s_l} c_r(\mathbf{d}_l)[\mathbf{z} \cdot \mathbf{d}_l] - \sum_{l \in N} \sum_{r \in s'_l} c_r(\mathbf{d}_l)[\mathbf{z} \cdot \mathbf{d}_l]
$$
$$
= \sum_{r \in s_i} c_r(\mathbf{d}_i)[\mathbf{z} \cdot \mathbf{d}_i] - \sum_{r \in s'_i} c_r(\mathbf{d}_i)[\mathbf{z} \cdot \mathbf{d}_i]
$$

[because only $i$'s strategy changed between $\mathbf{s}$ and $\mathbf{s}'$]

$$
= \sum_{r \in s_i \setminus s'_i} c_r(\mathbf{d}_i)[\mathbf{z} \cdot \mathbf{d}_i] - \sum_{r \in s'_i \setminus s_i} c_r(\mathbf{d}_i)[\mathbf{z} \cdot \mathbf{d}_i].
$$
$$
= \sum_{r \in s_i \setminus s'_i} \left( a_r[\mathbf{z} \cdot \mathbf{d}_i] + b_r \right)[\mathbf{z} \cdot \mathbf{d}_i] - \sum_{r \in s'_i \setminus s_i} \left( a_r[\mathbf{z} \cdot \mathbf{d}_i] + b_r \right)[\mathbf{z} \cdot \mathbf{d}_i]
$$
$$
= \sum_{r \in s_i \setminus s'_i} \left( a_r[\mathbf{z} \cdot \mathbf{d}_i]^2 + b_r[\mathbf{z} \cdot \mathbf{d}_i] \right) - \sum_{r \in s'_i \setminus s_i} \left( a_r[\mathbf{z} \cdot \mathbf{d}_i]^2 + b_r[\mathbf{z} \cdot \mathbf{d}_i] \right).
$$

Combining the differences in $\Phi_1$ and $\Phi_2$, following is the difference in the proposed potential function.

$$
\Phi(\mathbf{s}) - \Phi(\mathbf{s}')
$$
$$
= \Phi_1(\mathbf{s}) - \Phi_1(\mathbf{s}') + \Phi_2(\mathbf{s}) - \Phi_2(\mathbf{s}')
$$
$$
= \sum_{r \in s_i \setminus s'_i} \left( 2a_r[\mathbf{z} \cdot \mathbf{d}_i][\mathbf{z} \cdot \mathbf{x}_r(\mathbf{s})] + 2b_r[\mathbf{z} \cdot \mathbf{d}_i] \right) - \sum_{r \in s'_i \setminus s_i} \left( 2a_r[\mathbf{z} \cdot \mathbf{d}_i][\mathbf{z} \cdot \mathbf{x}_r(\mathbf{s})] + 2a_r[\mathbf{z} \cdot \mathbf{d}_i]^2 + 2b_r[\mathbf{z} \cdot \mathbf{d}_i] \right)
$$
$$
= \sum_{r \in s_i \setminus s'_i} 2[\mathbf{z} \cdot \mathbf{d}_i]\left( a_r[\mathbf{z} \cdot \mathbf{x}_r(\mathbf{s})] + b_r \right) - \sum_{r \in s'_i \setminus s_i} 2[\mathbf{z} \cdot \mathbf{d}_i]\left( a_r[\mathbf{z} \cdot \mathbf{x}_r(\mathbf{s})] + a_r[\mathbf{z} \cdot \mathbf{d}_i]) + b_r \right)
$$
$$
= \sum_{r \in s_i \setminus s'_i} 2[\mathbf{z} \cdot \mathbf{d}_i]\left( a_r[\mathbf{z} \cdot \mathbf{x}_r(\mathbf{s})] + b_r \right) - \sum_{r \in s'_i \setminus s_i} 2[\mathbf{z} \cdot \mathbf{d}_i]\left( a_r\left[\mathbf{z} \cdot (\mathbf{x}_r(\mathbf{s}) + \mathbf{d}_i)\right] + b_r \right)
$$
$$
= \sum_{r \in s_i \setminus s'_i} 2[\mathbf{z} \cdot \mathbf{d}_i]c_r(\mathbf{x}_r(\mathbf{s})) - \sum_{r \in s'_i \setminus s_i} 2[\mathbf{z} \cdot \mathbf{d}_i]c_r(\mathbf{x}_r(\mathbf{s}'))
$$
$$
= 2[\mathbf{z} \cdot \mathbf{d}_i]\left( \sum_{r \in s_i \setminus s'_i} c_r(\mathbf{x}_r(\mathbf{s})) - \sum_{r \in s'_i \setminus s_i} c_r(\mathbf{x}_r(\mathbf{s}')) \right).
$$

The difference between player $i$'s costs under $\mathbf{s}$ and $\mathbf{s}'$ is

$$
\pi_i(\mathbf{s}) - \pi_i(\mathbf{s}')
$$
$$
= \sum_{r \in s_i} c_r(\mathbf{x}_r(\mathbf{s})) - \sum_{r \in s'_i} c_r(\mathbf{x}_r(\mathbf{s}'))
$$
$$
= \sum_{r \in s_i \setminus s'_i} c_r(\mathbf{x}_r(\mathbf{s})) - \sum_{r \in s'_i \setminus s_i} c_r(\mathbf{x}_r(\mathbf{s}')) \text{ [by Eqn 4]}
$$

Therefore, for any player $i$ and any strategy profile $\mathbf{s}$ and $\mathbf{s}'$ (as defined above),

$$
\Phi(\mathbf{s}) - \Phi(\mathbf{s}') = 2[\mathbf{z} \cdot \mathbf{d}_i](\pi_i(\mathbf{s}) - \pi_i(\mathbf{s}')). \tag{6}
$$

If $[\mathbf{z} \cdot \mathbf{d}_i] > 0$ for all $i$, then this concludes the proof that multidimensional congestion games with a linear cost function are $\mathbf{w}$-potential games with $w_i = \frac{1}{2[\mathbf{z} \cdot \mathbf{d}_i]}$ for each $i$. However, if $[\mathbf{z} \cdot \mathbf{d}_i] = 0$ for some $i$, note that player $i$ does not affect the payoff of any other player. This is because $c_r(\mathbf{x}_r(\mathbf{s})) = a_r[\mathbf{z} \cdot \mathbf{x}_r(\mathbf{s})] + b_r = a_r\left[\mathbf{z} \cdot (\mathbf{x}_r(s_{-i}) + \mathbf{d}_i)\right] + b_r = a_r[\mathbf{z} \cdot \mathbf{x}_r(s_{-i})] + a_r[\mathbf{z} \cdot \mathbf{d}_i] + b_r = a_r[\mathbf{z} \cdot \mathbf{x}_r(s_{-i})] + b_r$. As a result, we can exclude such players $i$ from the game without impacting the other players' choices, and the resulting game is a $\mathbf{w}$-potential game.[2] $\qquad\square$

Below, we formalize the best-response algorithm outlined in the main text.

---

[2] For the purpose of equilibrium computation, the best responses of the excluded player $i$ can be added back later on without impacting the choices of the other players.

**Algorithm 2:** Best Response Dynamics

---

**Input:** A multidimensional congestion game

**Output:** Pure-strategy Nash equilibrium

1 Choose an arbitrary strategy profile $\mathbf{s}$
2 **while** *some player $i$ can improve their cost* **do**
3 $\quad$ Update $s_i$ with $i$'s best response to $\mathbf{s}_{-i}$
4 **end**
5 return $\mathbf{s}$

---

To analyze the best-response algorithm, we establish an upper bound on the potential function in Appendix Lemma 3. We make the typical integrality assumption on all $a_r$, $b_r$ and the elements of the vectors $\mathbf{z}$ and $\mathbf{d}_i$ for any player $i$ [Fotakis et al., 2002, 2005]. We use the below notations. The sum of all players' demand vectors is denoted by $\mathbf{d}_N \equiv \sum_{i \in N} \mathbf{d}_i$. In addition, let $A \equiv \sum_{r \in R} a_r$ and $B \equiv \sum_{r \in R} b_r$.

**Appendix Lemma 3.** *For any strategy profile $\mathbf{s}$, the potential function $\Phi(\mathbf{s})$ is upper bounded by $2A[\mathbf{z} \cdot \mathbf{d}_N]^2 + (n+1)B[\mathbf{z} \cdot \mathbf{d}_N]$.*

*Proof.* We first get the following bounds on the $\Phi_1$ and $\Phi_2$ functions.

$$
\begin{aligned}
\Phi_1(\mathbf{s}) &= \sum_{r \in R} c_r(\mathbf{x}_r(\mathbf{s}))[\mathbf{z} \cdot \mathbf{x}_r(\mathbf{s})] \\
&\leq \sum_{r \in R} c_r(\mathbf{x}_r(\mathbf{s}))[\mathbf{z} \cdot \mathbf{d}_N] \\
&= [\mathbf{z} \cdot \mathbf{d}_N] \sum_{r \in R} c_r(\mathbf{x}_r(\mathbf{s})) \\
&\leq [\mathbf{z} \cdot \mathbf{d}_N] \sum_{r \in R} c_r(\mathbf{d}_N).
\end{aligned}
$$

$$
\begin{aligned}
\Phi_2(\mathbf{s}) &= \sum_{i \in N} \sum_{r \in s_i} c_r(\mathbf{d}_i)[\mathbf{z} \cdot \mathbf{d}_i] \\
&\leq \sum_{r \in R} \sum_{i \in N} c_r(\mathbf{d}_i)[\mathbf{z} \cdot \mathbf{d}_i] \\
&= \sum_{i \in N} [\mathbf{z} \cdot \mathbf{d}_i] \sum_{r \in R} c_r(\mathbf{d}_i) \\
&\leq [\mathbf{z} \cdot \mathbf{d}_N] \sum_{i \in N} \sum_{r \in R} c_r(\mathbf{d}_i) \\
&= [\mathbf{z} \cdot \mathbf{d}_N] \sum_{i \in N} \sum_{r \in R} \left( a_r[\mathbf{z} \cdot \mathbf{d}_i] + b_r \right) \\
&= [\mathbf{z} \cdot \mathbf{d}_N] \sum_{r \in R} \left( a_r[\mathbf{z} \cdot \mathbf{d}_N] + nb_r \right) \\
&= [\mathbf{z} \cdot \mathbf{d}_N] \left( \sum_{r \in R} \left( a_r[\mathbf{z} \cdot \mathbf{d}_N] + b_r \right) + (n-1) \sum_{r \in R} b_r \right) \\
&= [\mathbf{z} \cdot \mathbf{d}_N] \left( \sum_{r \in R} c_r(\mathbf{d}_N) \right) + (n-1)B[\mathbf{z} \cdot \mathbf{d}_N].
\end{aligned}
$$

Combining the bounds on $\Phi_1$ and $\Phi_2$, we get the following bound on the potential function.

$$\Phi(\mathbf{s}) \leq 2[\mathbf{z} \cdot \mathbf{d}_N]\left(\sum_{r \in R} c_r(\mathbf{d}_N)\right) + (n-1)B[\mathbf{z} \cdot \mathbf{d}_N]$$

$$= 2[\mathbf{z} \cdot \mathbf{d}_N]\left(\sum_{r \in R}\Big(a_r[z.\mathbf{d}_N] + b_r\Big)\right)$$

$$+ (n-1)B[\mathbf{z} \cdot \mathbf{d}_N]$$

$$= 2[\mathbf{z} \cdot \mathbf{d}_N]\Big(A[z.\mathbf{d}_N] + B\Big) + (n-1)B[\mathbf{z} \cdot \mathbf{d}_N]$$

$$= 2A[\mathbf{z} \cdot \mathbf{d}_N]^2 + (n+1)B[\mathbf{z} \cdot \mathbf{d}_N]$$

$\square$

Next, we upper bound the number of iterations. Each iteration runs in $\mathcal{O}\big(nkpm^2\big)$ time, giving us the following theorems.

**Appendix Theorem 4.** *The best-response algorithm runs in pseudo-polynomial time.*

*Proof.* Using Equation 6, whenever a player $i$ reduces their cost by 1 in Algorithm 2, the potential function decreases by $2[\mathbf{z} \cdot \mathbf{d}_i] \geq 2$ due to the integrality of $z$ and $\mathbf{d}_i$. Also, as detailed in the proof of Appendix Theorem 2, if $[\mathbf{z} \cdot \mathbf{d}_i] = 0$ for some players $i$, those players do not impact the cost of the other players and thereby can be excluded from the game. As a result, the maximum number of iterations of the algorithm is $A[\mathbf{z} \cdot \mathbf{d}_N]^2 + \frac{n+1}{2}B[\mathbf{z} \cdot \mathbf{d}_N]$. $\square$

The following theorem gives us the multidimensional counterpart of the single-dimensional result by Panagopoulou and Spirakis [2007].

**Theorem 10.** *For linear-cost $k$-DCGs, the best-response algorithm runs in polynomial time if $\max_r a_r, \max_r b_r$, and $\frac{\max_i[\mathbf{z}\cdot\mathbf{d}_i]^2}{\min_i[\mathbf{z}\cdot\mathbf{d}_i]}$ are polynomial in $n$.*

*Proof.* Whenever a player $i$ reduces their cost by 1, the decrease in the potential function is $2[\mathbf{z} \cdot \mathbf{d}_i] \geq 2\min_i[\mathbf{z} \cdot \mathbf{d}_i]$.[3]

Number of iterations

$$= \frac{2A[\mathbf{z} \cdot \mathbf{d}_N]^2}{2\min_i[\mathbf{z} \cdot \mathbf{d}_i]} + \frac{(n+1)B}{2}\frac{[\mathbf{z} \cdot \mathbf{d}_N]}{\min_i[\mathbf{z} \cdot \mathbf{d}_i]}$$

$$\leq \frac{A\Big(n^2 \max_i[\mathbf{z} \cdot \mathbf{d}_i]^2\Big)}{\min_i[\mathbf{z} \cdot \mathbf{d}_i]} + \frac{(n+1)B}{2}\frac{n\max_i[\mathbf{z} \cdot \mathbf{d}_i]}{\min_i[\mathbf{z} \cdot \mathbf{d}_i]}$$

$$\leq n^2(A+B)\frac{\max_i[\mathbf{z} \cdot \mathbf{d}_i]^2}{\min_i[\mathbf{z} \cdot \mathbf{d}_i]}$$

$$\leq n^2 m(\max_r a_r + \max_r b_r)\frac{\max_i[\mathbf{z} \cdot \mathbf{d}_i]^2}{\min_i[\mathbf{z} \cdot \mathbf{d}_i]}.$$

Therefore, when $\max_r a_r, \max_r b_r$, and $\frac{\max_i[\mathbf{z}\cdot\mathbf{d}_i]^2}{\min_i[\mathbf{z}\cdot\mathbf{d}_i]}$ are polynomial in $n$, then the number of iterations is $\mathcal{O}\big(m \times \text{poly}(n)\big)$, where $n$ is the number of players and $m$ is the number of resources, as defined earlier. $\square$

We next consider the cases of *binary demand vectors* and $k$-*CCGs*.

**Theorem 11.** *For linear-cost $k$-DCGs with binary demand vectors vectors, the best-response algorithm runs in polynomial time if the following cost function parameters are polynomial in $n$: $\max_r a_r$, $\max_r b_r$, and $\max_j z_j$.*

---

[3]Note that unlike one-dimensional demands, $\min_i \mathbf{d}_i$ is not well defined.

*Proof.* First, note that $\max_i[\mathbf{z} \cdot \mathbf{d}_i] \le \sum_{j \in [k]} z_j \le k \max_{j \in [k]} z_j$. Whenever a player $i$ reduces their cost by 1, the decrease in the potential function is $2[\mathbf{z} \cdot \mathbf{d}_i] \ge 2\min_i[\mathbf{z} \cdot \mathbf{d}_i] \ge 2$.

Number of iterations

$$
\begin{aligned}
&= \frac{2A[\mathbf{z} \cdot \mathbf{d}_N]^2}{2} + \frac{(n+1)B[\mathbf{z} \cdot \mathbf{d}_N]}{2} \\
&\le An^2 \max_i[\mathbf{z} \cdot \mathbf{d}_i]^2 + \frac{(n+1)B}{2} n \max_i[\mathbf{z} \cdot \mathbf{d}_i] \\
&\le n^2(A+B) \max_i[\mathbf{z} \cdot \mathbf{d}_i]^2 \\
&\le n^2 m(\max_r a_r + \max_r b_r)\big(k \max_j z_j\big)^2.
\end{aligned}
$$

Therefore, the number of iterations is $\mathcal{O}(mk^2 \times \text{poly}(n))$. $\square$

**Theorem 12.** *For linear-cost $k$-CCGs, the best-response algorithm runs in polynomial time if $\max_r a_r$, $\max_r b_r$, $\frac{\max_j z_j^2}{\min_j z_j}$, and $\frac{\max_i d_{i,l(i)}^2}{\min_i d_{i,l(i)}}$ are polynomial in $n$, where $l(i) \in [k]$ denotes the index of the non-zero element in $\mathbf{d}_i$.*

*Proof.* We get $\max_i[\mathbf{z} \cdot \mathbf{d}_i] = \max_i \big[z_{l(i)} d_{i,l(i)}\big] \le \big(\max_i z_{l(i)}\big)\big(\max_i d_{i,l(i)}\big) \le \big(\max_j z_j\big)\big(\max_i d_{i,l(i)}\big)$.
Similarly, $\min_i[\mathbf{z} \cdot \mathbf{d}_i] \ge \big(\min_j z_j\big)\big(\min_i d_{i,l(i)}\big)$.
Using Theorem 10, the number of iterations is at most
$n^2 m(\max_r a_r + \max_r b_r)\frac{\max_j z_j^2}{\min_j z_j}\frac{\max_i d_{i,l(i)}^2}{\min_i d_{i,l(i)}}$.

Number of iterations

$$
\le n^2 m(\max_r a_r + \max_r b_r)\frac{\max_j z_j^2}{\min_j z_j}\frac{\max_i d_{i,l(i)}^2}{\min_i d_{i,l(i)}}.
$$

Therefore, the number of iterations is $\mathcal{O}(n^2 m \times \text{poly}(n))$. $\square$

**Experimental Results.** We have performed experiments to evaluate the performance of Algorithm 2. The cost function is $c_r(\mathbf{x}_r(\mathbf{s})) = a_r[\mathbf{z} \cdot \mathbf{x}_r(\mathbf{s})] + b_r$, where $a_r$ and $b_r$ are random integers between 0 and 5, and $\mathbf{z}$ is a vector of random integers between 0 and 5 (all inclusive). Every player has a random demand vector where each element is between 0 and 5 (both inclusive except that the demand vector cannot be all zeros). In the implementation of the algorithm, we order the players $i$ from the highest to lowest value of $\mathbf{z} \cdot \mathbf{d}_i$.

We vary the number of dimensions $k = 2, 3, 4$ and the number of links from 2 to 10. We also vary the number of players from 5 to 100. Figure 4 illustrates our experimental results for a subset of representative experiments. It shows that Algorithm 2 scales up gracefully as we increase the number of players and links. This is perhaps not surprising given the pseudopolynomial running time of the algorithm. Our experiments are consistent with those on single-dimensional weighted congestion games [Panagopoulou and Spirakis, 2007].

## EXPONENTIAL COST FUNCTIONS

We know that $k$-DCGs with cost functions of the shape $c_r(\mathbf{x}_r(\mathbf{s})) = a_r \exp(\mathbf{z} \cdot \mathbf{x}_r(\mathbf{s})) + b_r$ are isomorphic to one-dimensional congestion games [Klimm and Schütz, 2022]. We use Harks et al. [2011]'s results on 1-DCGs to derive the following potential function for $k$-DCGs.

**Appendix Theorem 5.** *Any multidimensional congestion game with exponential resource costs is a $\mathbf{w}$-potential game.*

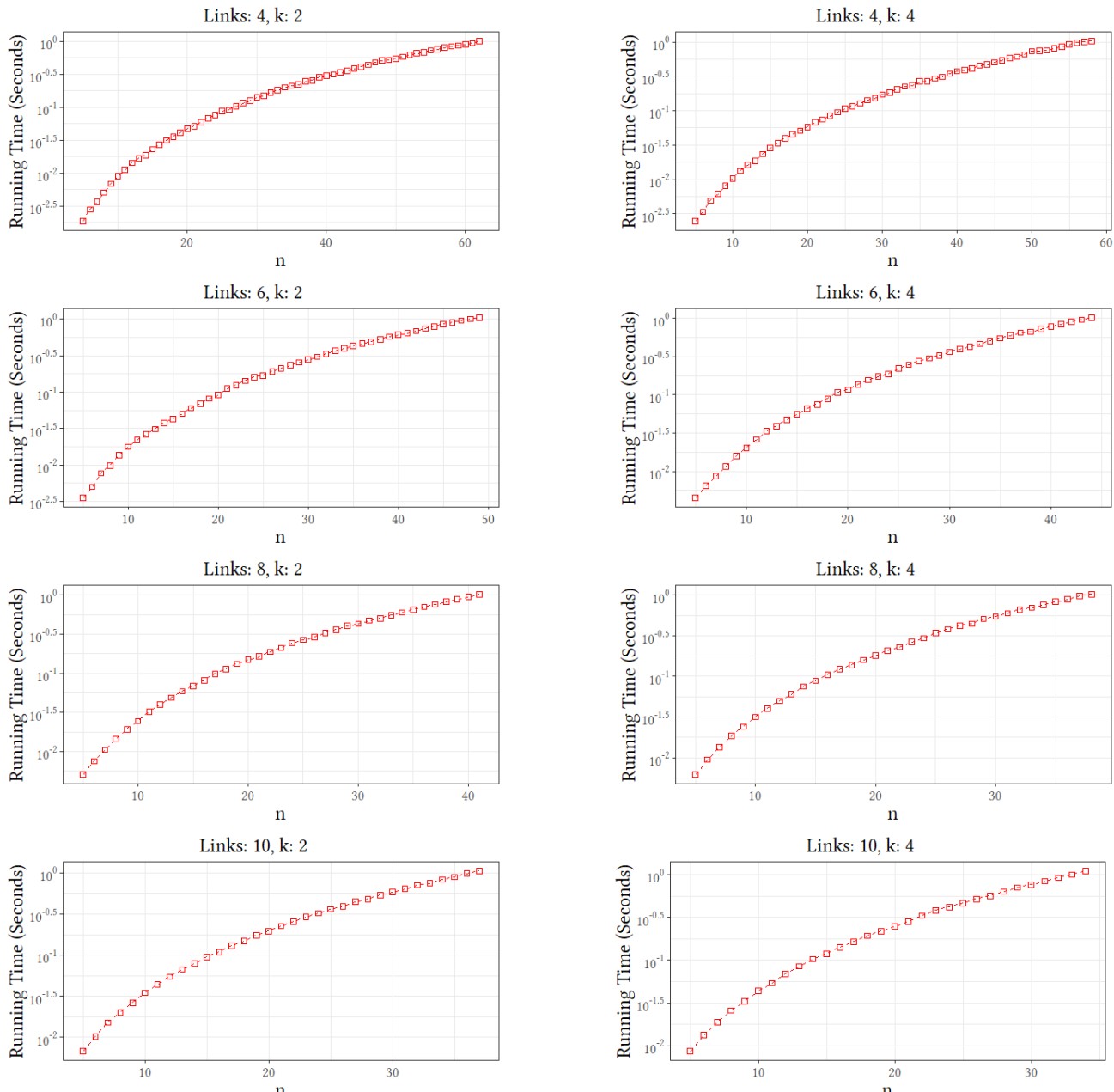

Figure 4: Performance of the learning dynamics algorithm for linear cost functions. The figures show that Algorithm 2 scales up nicely with an increasing number of links and dimension $k$. Instead of selecting the players in a linear order, we prioritize players with higher weights according to $\mathbf{z} \cdot \mathbf{d}_i$ for player $i$. This leads to a greater impact on the potential function.

*Proof.* We show that the $\Phi(\mathbf{s})$ defined below is a $\mathbf{w}$-potential function for the choice of $w_i = \frac{1}{1-\exp(-\mathbf{z}\cdot\mathbf{d}_i)}$ for each $i$.

$$\Phi_1(\mathbf{s}) = \sum_{r\in R} c_r(\mathbf{x}_r(\mathbf{s})).$$

$$\Phi_2(\mathbf{s}) = \sum_{i\in N}\sum_{r\in s_i} b_r(1 - \exp(-\mathbf{z}\cdot\mathbf{d}_i)).$$

$$\Phi(\mathbf{s}) = \Phi_1(\mathbf{s}) + \Phi_2(\mathbf{s}). \tag{7}$$

Consider any set of resources $s_i' \neq s_i$. Define $\mathbf{s}' = (\mathbf{s}_{-i}, s_i')$. For any resource $r$ that is picked either by both $s_i$ and $s_i'$ or none of them:

$$c_r(\mathbf{x}_r(\mathbf{s})) = c_r(\mathbf{x}_r(\mathbf{s}')). \tag{8}$$

For any $r \in s_i \setminus s_i'$:

$$
\begin{aligned}
&c_r(\mathbf{x}_r(\mathbf{s})) - c_r(\mathbf{x}_r(\mathbf{s}'))\\
&= a_r \exp(\mathbf{z}\cdot\mathbf{x}_r(\mathbf{s})) + b_r - a_r \exp(\mathbf{z}\cdot\mathbf{x}_r(\mathbf{s}')) - b_r\\
&= a_r \exp(\mathbf{z}\cdot\mathbf{x}_r(\mathbf{s})) - a_r \exp(\mathbf{z}\cdot(\mathbf{x}_r(\mathbf{s}) - \mathbf{d}_i))\\
&= a_r \exp(\mathbf{z}\cdot\mathbf{x}_r(\mathbf{s})) - a_r \exp(\mathbf{z}\cdot\mathbf{x}_r(\mathbf{s}))\exp(-\mathbf{z}\cdot\mathbf{d}_i)\\
&= a_r \exp(\mathbf{z}\cdot\mathbf{x}_r(\mathbf{s}))(1 - \exp(-\mathbf{z}\cdot\mathbf{d}_i)).
\end{aligned}
$$

Similarly, for any resource $r \in s_i' \setminus s_i$,

$$
\begin{aligned}
&c_r(\mathbf{x}_r(\mathbf{s})) - c_r(\mathbf{x}_r(\mathbf{s}'))\\
&= a_r \exp(\mathbf{z}\cdot(\mathbf{x}_r(\mathbf{s}') - \mathbf{d}_i)) + b_r - a_r \exp(\mathbf{z}\cdot\mathbf{x}_r(\mathbf{s})) - b_r\\
&= a_r \exp(\mathbf{z}\cdot\mathbf{x}_r(\mathbf{s}'))\exp(-\mathbf{z}\cdot\mathbf{d}_i) - a_r \exp(\mathbf{z}\cdot\mathbf{x}_r(\mathbf{s}))\\
&= -a_r \exp(\mathbf{z}\cdot\mathbf{x}_r(\mathbf{s}'))(1 - \exp(-\mathbf{z}\cdot\mathbf{d}_i)).
\end{aligned}
$$

The difference in the $\Phi_1$ function under $\mathbf{s}$ and $\mathbf{s}'$ is

$$
\begin{aligned}
&\Phi_1(\mathbf{s}) - \Phi_1(\mathbf{s}')\\
&= \sum_{r\in R}\Big(c_r(\mathbf{x}_r(\mathbf{s})) - c_r(\mathbf{x}_r(\mathbf{s}'))\Big)\\
&= \sum_{r\in s_i\setminus s_i'}\Big(c_r(\mathbf{x}_r(\mathbf{s})) - c_r(\mathbf{x}_r(\mathbf{s}'))\Big) + \sum_{r\in s_i'\setminus s_i}\Big(c_r(\mathbf{x}_r(\mathbf{s})) - c_r(\mathbf{x}_r(\mathbf{s}'))\Big)\\
&= \sum_{r\in s_i\setminus s_i'}\Big(a_r \exp(\mathbf{z}\cdot\mathbf{x}_r(\mathbf{s}))(1 - \exp(-\mathbf{z}\cdot\mathbf{d}_i))\Big) + \sum_{r\in s_i'\setminus s_i}\Big(-a_r \exp(\mathbf{z}\cdot\mathbf{x}_r(\mathbf{s}'))(1 - \exp(-\mathbf{z}\cdot\mathbf{d}_i))\Big)\\
&= (1 - \exp(-\mathbf{z}\cdot\mathbf{d}_i))\left(\sum_{r\in s_i\setminus s_i'} a_r \exp(\mathbf{z}\cdot\mathbf{x}_r(\mathbf{s})) - \sum_{r\in s_i'\setminus s_i} a_r \exp(\mathbf{z}\cdot\mathbf{x}_r(\mathbf{s}'))\right).
\end{aligned}
$$

The difference in the $\Phi_2$ function under $\mathbf{s}$ and $\mathbf{s}'$ is

$$
\begin{aligned}
&\Phi_2(\mathbf{s}) - \Phi_2(\mathbf{s}')\\
&= \sum_{l\in N}\sum_{r\in s_l} b_r(1 - \exp(-\mathbf{z}\cdot\mathbf{d}_l)) - \sum_{l\in N}\sum_{r\in s_l'} b_r(1 - \exp(-\mathbf{z}\cdot\mathbf{d}_l))\\
&= \sum_{r\in s_i} b_r(1 - \exp(-\mathbf{z}\cdot\mathbf{d}_i)) - \sum_{r\in s_i'} b_r(1 - \exp(-\mathbf{z}\cdot\mathbf{d}_i))\\
&\qquad\text{[because only } i\text{'s strategy changed between } \mathbf{s} \text{ and } \mathbf{s}']\\
&= \sum_{r\in s_i\setminus s_i'} b_r(1 - \exp(-\mathbf{z}\cdot\mathbf{d}_i)) - \sum_{r\in s_i'\setminus s_i} b_r(1 - \exp(-\mathbf{z}\cdot\mathbf{d}_i))\\
&= (1 - \exp(-\mathbf{z}\cdot\mathbf{d}_i))\left(\sum_{r\in s_i\setminus s_i'} b_r - \sum_{r\in s_i'\setminus s_i} b_r\right).
\end{aligned}
$$

Combining the differences in $\Phi_1$ and $\Phi_2$, following is the difference in the proposed potential function.

$$
\begin{aligned}
\Phi(\mathbf{s}) &- \Phi(\mathbf{s}') \\
&= \Phi_1(\mathbf{s}) - \Phi_1(\mathbf{s}') + \Phi_2(\mathbf{s}) - \Phi_2(\mathbf{s}') \\
&= \big(1 - \exp(-\mathbf{z} \cdot \mathbf{d}_i)\big) \Bigg( \sum_{r \in s_i \setminus s_i'} \Big( a_r \exp(\mathbf{z} \cdot \mathbf{x}_r(\mathbf{s})) + b_r \Big) - \sum_{r \in s_i' \setminus s_i} \Big( a_r \exp(\mathbf{z} \cdot \mathbf{x}_r(\mathbf{s}')) + b_r \Big) \Bigg) \\
&= \big(1 - \exp(-\mathbf{z} \cdot \mathbf{d}_i)\big) \Bigg( \sum_{r \in s_i \setminus s_i'} c_r(\mathbf{x}_r(\mathbf{s})) - \sum_{r \in s_i' \setminus s_i} c_r(\mathbf{x}_r(\mathbf{s}')) \Bigg) \\
&= \big(1 - \exp(-\mathbf{z} \cdot \mathbf{d}_i)\big) \Bigg( \sum_{r \in s_i} c_r(\mathbf{x}_r(\mathbf{s})) - \sum_{r \in s_i'} c_r(\mathbf{x}_r(\mathbf{s}')) \Bigg) \\
&\qquad\qquad\qquad\qquad\qquad\qquad \text{[by Eqn 8]} \\
&= \big(1 - \exp(-\mathbf{z} \cdot \mathbf{d}_i)\big) \big(\pi_i(\mathbf{s}) - \pi_i(\mathbf{s}')\big).
\end{aligned}
$$

Therefore,

$$
\Phi(\mathbf{s}) - \Phi(\mathbf{s}') = \big(1 - \exp(-\mathbf{z} \cdot \mathbf{d}_i)\big) \big(\pi_i(\mathbf{s}) - \pi_i(\mathbf{s}')\big). \tag{9}
$$

$\square$

We next give an upper bound on the potential function defined in Appendix Theorem 5. As defined in the previous subsection, $\mathbf{d}_N = \sum_{i \in N} \mathbf{d}_i$.

**Appendix Lemma 6.** *The potential function for multidimensional congestion games with an exponential cost function is upper bounded by $m \exp(\mathbf{z} \cdot \mathbf{d}_N) \max_r a_r + (n+1) m \max_r b_r$.*

*Proof.* We get

$$
\begin{aligned}
\Phi(\mathbf{s}) &= \sum_{r \in R} c_r(\mathbf{x}_r(\mathbf{s})) + \sum_{i \in N} \sum_{r \in s_i} b_r(1 - \exp(-\mathbf{z} \cdot \mathbf{d}_i)) \\
&= \sum_{r \in R} \big( a_r \exp(\mathbf{z} \cdot \mathbf{x}_r(\mathbf{s})) + b_r \big) + \sum_{i \in N} \sum_{r \in s_i} b_r(1 - \exp(-\mathbf{z} \cdot \mathbf{d}_i)) \\
&\leq \sum_{r \in R} \big( a_r \exp(\mathbf{z} \cdot \mathbf{d}_N) + b_r \big) + \sum_{i \in N} \sum_{r \in R} b_r(1 - 0) \\
&\leq \exp(\mathbf{z} \cdot \mathbf{d}_N) \sum_{r \in R} a_r + \sum_{r \in R} b_r + \sum_{i \in N} \Big( m \max_r b_r \Big) \\
&\leq \exp(\mathbf{z} \cdot \mathbf{d}_N) m \max_r a_r + m \max_r b_r + n m \max_r b_r \\
&= m \exp(\mathbf{z} \cdot \mathbf{d}_N) \max_r a_r + (n+1) m \max_r b_r.
\end{aligned}
$$

$\square$

Following is the running time analysis of the best response algorithm. Recall that each iteration runs in $\mathcal{O}\big(nkpm^2\big)$ time.

**Theorem 13.** *The best-response algorithm runs in polynomial time for exponential-cost $k$-DCGs if $\max_r a_r$ and $\max_r b_r$ are polynomial in $n$ and $[\mathbf{z} \cdot \mathbf{d}_N]$ is $\mathcal{O}(\log n)$.*

*Proof.* Using Appendix Theorem 5, whenever a player $i$ reduces its cost by 1, the potential function reduces by $1 - \exp(-\mathbf{z} \cdot \mathbf{d}_i) \geq 1 - \frac{1}{e} = \frac{e-1}{e}$. Using Appendix Lemma 6, the number of iterations is at most $\frac{e}{e-1}\Big( m \exp(\mathbf{z} \cdot \mathbf{d}_N) \max_r a_r + (n+1) m \max_r b_r \Big)$.

$\square$

## APPROXIMATE PSNE FOR GENERAL COST FUNCTIONS

In this section, we remove the condition of monotonicity imposed in [Christodoulou et al., 2023] and give an $(\alpha, \beta)$-PSNE algorithm for arbitrary cost functions. For this, we first define a term that bounds the degree of non-monotonicity of the congestion functions.

Recall that $\mathbf{x}_r(\mathbf{s}) = \sum_{i \in N; r \in s_i} \mathbf{d}_i$ for any $\mathbf{s} \in S$. Let $\Delta_r = \max\{\max_{i \in N, \mathbf{s} \in S; r \in s_i} c_r(\mathbf{x}_r(\mathbf{s}) - \mathbf{d}_i) - c_r(\mathbf{x}_r(\mathbf{s})), 0\}$ be the maximum non-negative marginal decrease of any player under the cost function for resource $r \in R$. When the congestion function is nondecreasing, $\Delta_r = 0$. Otherwise, $\Delta_r > 0$. Let $\Delta_{\max} = \max_{r \in R} \Delta_r$. We obtain the following result that generalizes the result in [Christodoulou et al., 2023].

**Theorem 14.** *Every $k$-DCG has an $(\alpha, \beta)$-PSNE for $\alpha = n$ and $\beta = (n-1)m\Delta_{\max}$. Furthermore, it can be computed using an iterative algorithm that is guaranteed to converge.*

*Proof.* Following the idea from Christodoulou et al. [2023], we start by providing a bound for the change of other player costs when a player $i$ changes its strategies. For any strategy profile $\mathbf{s} = (s_1, ..., s_n) \in S$, $s_i \neq s_i' \in S_i$, and $i \neq l \in N$, we have that

$$
\begin{aligned}
&\pi_l(s_i', \mathbf{s}_{-i}) - \pi_l(s_i, \mathbf{s}_{-i}) \\
&= \sum_{r \in s_l} c_r(\mathbf{x}_r(s_i', \mathbf{s}_{-i})) - \sum_{r \in s_l} c_r(\mathbf{x}_r(s_i, \mathbf{s}_{-i})) \\
&= \sum_{r \in s_l \cap (s_i' \setminus s_i)} c_r(\mathbf{x}_r(s_i', \mathbf{s}_{-i})) - c_r(\mathbf{x}_r(s_i, \mathbf{s}_{-i})) + \sum_{r \in s_l \cap (s_i \setminus s_i')} c_r(\mathbf{x}_r(s_i', \mathbf{s}_{-i})) - c_r(\mathbf{x}_r(s_i, \mathbf{s}_{-i})) \\
&\leq \sum_{r \in s_l \cap (s_i' \setminus s_i)} c_r(\mathbf{x}_r(s_i', \mathbf{s}_{-i})) - c_r(\mathbf{x}_r(s_i, \mathbf{s}_{-i})) + \Delta_{\max} |s_l \cap (s_i \setminus s_i')| \\
&\leq \sum_{r \in s_i'} c_r(\mathbf{x}_r(s_i', \mathbf{s}_{-i})) + m\Delta_{\max} \\
&= \pi_i(s_i', \mathbf{s}_{-i}) + m\Delta_{\max}.
\end{aligned}
$$

Above, the first equality is by the definition of player cost functions, the second equality is by removing terms that $s_i'$ do not affect and splitting terms into those that increase or decrease the total weights, the third inequality is noting that the change of each of the second summarization terms is bounded by $\Delta_{\max}$, and the fourth inequality is by dropping the subtracted terms.

Define $\Pi(\mathbf{s}) = \sum_{i \in N} \pi_i(\mathbf{s})$ to be the social cost of the players under $\mathbf{s}$. By summing up all of the inequalities above except player $i \in N$, we have that

$$
\begin{aligned}
&\sum_{l \neq i \in N} \left( \pi_l(s_i', \mathbf{s}_{-i}) - \pi_l(s_i, \mathbf{s}_{-i}) \right) \leq (n-1)[\pi_i(s_i', \mathbf{s}_{-i}) + m\Delta_{\max}] \\
&(\Pi(s_i', \mathbf{s}_{-i}) - \pi_i(s_i', \mathbf{s}_{-i})) - (\Pi(s_i, \mathbf{s}_{-i}) - \pi_i(s_i, \mathbf{s}_{-i})) \leq (n-1)[\pi_i(s_i', \mathbf{s}_{-i}) + m\Delta_{\max}] \\
&\Pi(s_i', \mathbf{s}_{-i}) - \Pi(s_i, \mathbf{s}_{-i}) \leq n\pi_i(s_i', \mathbf{s}_{-i}) - \pi_i(s_i, \mathbf{s}_{-i}) + (n-1)m\Delta_{\max}.
\end{aligned}
$$

If $\pi_i(s_i, \mathbf{s}_{-i}) > n\pi_i(s_i', \mathbf{s}_{-i}) + (n-1)m\Delta_{\max}$, then the social cost of the players must strictly decrease by deviating to $s_i'$. Because the social cost has a local minima, it follows that any $\mathbf{s}^{opt} \in S$ that minimizes $\Pi$ is an $(\alpha, \beta)$-approximate PSNE for $\alpha = n$ and $\beta = (n-1)m\Delta_{\max}$.

We can compute an $(\alpha, \beta)$-PSNE using an iterative procedure where at each round, if $\pi_i(s_i, \mathbf{s}_{-i}) > n\pi_i(s_i', \mathbf{s}_{-i}) + (n-1)m\Delta_{\max}$ holds for any player $i$ currently playing $s_i$, the player deviates to $s_i'$. As the set of strategy profiles is finite, we eventually reach an $(\alpha, \beta)$-PSNE. $\qquad \square$

## F  STRUCTURED COSTS AND DEMANDS

Our study of structured costs and demands is motivated by a variety of realistic examples of traffic congestion games, where resources represent roads. As an example of structured/ordered demands, vehicles can be ordered by their demand vectors

representing width, length, weight, etc. (e.g., semis, pickup trucks, SUVs, sedans, and so on). A common example of a nondecreasing cost function is more vehicles on the road means higher costs for everyone. Singleton strategies are seen in grid-patterned road networks with parallel roads to go from source to destination [Milchtaich, 2006]. We also consider structured cost functions– e.g., different types of roads have different speed limits: highways, county routes, local roads, etc.

## ORDERED DEMAND, NONDECREASING COST, AND SINGLETON STRATEGIES

Suppose that the players can be ordered according to their demand vectors: $\mathbf{d}_1 \geq \mathbf{d}_2 \geq ... \geq \mathbf{d}_n$ (w.l.o.g.). Let each player $i$'s set of *singleton* strategies $S_i = \{\{r\} \mid r \in R\}$. In addition, assume that the cost functions are nondecreasing. We can compute a PSNE using the greedy best response algorithm, which orders the players from high to low demand and lets them play their best response in that order [Milchtaich, 2006]. Details are in the Appendix.

**Theorem 15.** *For a $k$-DCG with ordered demand vectors, nondecreasing cost functions, and singleton-resource strategies, a PSNE can be computed in $\mathcal{O}(n \log n + nmk)$ time.*

*Proof.* We sort and iterate through the players in the order of high to low demand vectors: 1, 2, ..., $n$ (w.l.o.g.). Sorting takes $\mathcal{O}(n \log n)$. At each iteration, a player $j$ chooses the best-response strategy with respect to the choices of the previous players. None of the previous players $i$ would have any incentive to deviate because $\mathbf{d}_i \geq \mathbf{d_j}$ and the cost functions are nondecreasing. That is, if a previous player $i$ could benefit from deviating to $r$, the current player $j$ would have chosen $r$. By keeping track of the aggregate demand vector for each resource, we get the result. $\square$

## ORDERED DEMAND, NONDECREASING COST, AND SHARED STRATEGIES

We relax the assumption of singleton-resource strategies. We show that as long as the players have the same set of strategies, we can compute a PSNE efficiently using the greedy best response algorithm.

**Theorem 16.** *For a $k$-DCG with ordered demand vectors, nondecreasing cost functions, and a shared set of strategies of size $p$, a PSNE can be computed in $\mathcal{O}(n \log n + npmk)$.*

*Proof Sketch.* The proof of Theorem 15 extends from singleton resources to sets of resources because the cost functions are additive over the resources. $\square$

## STRUCTURED COST FUNCTIONS AND SINGLETON STRATEGIES

In this scenario, we do not assume any ordering among the demands of the players. Instead, we assume that the cost functions are nondecreasing and that the resources are ordered by their cost functions. That is, w.l.o.g., $c_1(\mathbf{x}) \geq c_2(\mathbf{x}) \geq ... \geq c_m(\mathbf{x})$ for any $\mathbf{x}$. We also assume that there are constants $\alpha_j \geq 1$ such that $c_{j-1}(\mathbf{x}) = \alpha_j c_j(\mathbf{x})$ for any resource $j > 1$ and $\mathbf{x}$. These assumptions mean that some resources are more costly than others and that the costs of the resources are "nicely separated." Finally, we assume singleton-resource strategies. We get the following result.

**Theorem 17.** *For a $k$-DCG with nondecreasing and structured cost functions, where there are constants $\alpha_j \geq 1$ such that $c_{j-1}(\mathbf{x}) = \alpha_j c_j(\mathbf{x})$ for any resource $j > 1$ and aggregate demand vector $\mathbf{x}$, and singleton-resource strategies, a PSNE can be computed in $\mathcal{O}(n \log n + nmk)$ time.*

*Proof Sketch.* We can compute a PSNE in such $k$-DCGs using the greedy best response algorithm. We first order the players according to the cost $c_1$ of their demand vectors. W.l.o.g., let $c_1(\mathbf{d}_1) \geq c_1(\mathbf{d}_2) \geq ... \geq c_1(\mathbf{d}_n)$. Note that we are *not* assuming $\mathbf{d}_1 \geq \mathbf{d}_2 \geq ... \geq \mathbf{d}_n$. In fact, the demand vectors may not be comparable at all. We next prove by induction that the same ordering of players (1, 2, ... $n$) applies to the cost function of every resource. Suppose this is true for resource $j - 1$. We show it to be true for resource $j$. Consider any two consecutive players $i - 1$ and $i$. By assumptions, $c_{j-1}(\mathbf{d}_{i-1}) \geq c_{j-1}(\mathbf{d}_i)$, $c_{j-1}(\mathbf{d}_{i-1}) = \alpha_j c_j(\mathbf{d}_{i-1})$, and $c_{j-1}(\mathbf{d}_i) = \alpha_j c_j(\mathbf{d}_i)$. Therefore, $c_j(\mathbf{d}_{i-1}) \geq c_j(\mathbf{d}_i)$.

Therefore, even though we cannot order the demand vectors intrinsically, we are able to order them w.r.t. the cost functions, which is all that matters for greedy best response. $\square$