# OpenReview forum: "Equilibrium Computation in Multidimensional Congestion Games: CSP and Learning Dynamics Approaches"
_auai.org/UAI/2024/Conference — UAI 2024 poster_

### Official Review · Reviewer_avXR · 2024-03-19

**Q2-1 Originality-Novelty:** 2
**Q2-2 Correctness-Technical Quality:** 3
**Q2-5 Clarity Of Writing:** 2

**Q1 Summary And Contributions:**

This paper mainly presents (algorithmic) complexity results about several variations of multidimensional congestion games (the question being to find a pure Nash equilibrium);   the  algorithmic approach is  based on a modelling of the problem by a CSP, solved in its dual version.
Experiments are presented that compare several implementations of the approach to each other, but do not include a comparison to state of the art algorithms.

**Q2-3 Extent To Which Claims Are Supported By Evidence:**

2: Fair: the main claims are somewhat supported by evidence (but the experimental evaluation may be weak, or does not match entirely with the claims, important baselines may be missing, proofs contain important ideas but lack rigor, algorithmic details are only discussed superficially, references are imprecise, assumptions are not sufficiently motivated or explicated, etc.).

**Q2-4 Reproducibility:**

3: Good: key resources (e.g. proofs, code, data) are available and key details (e.g. proofs, experimental setup) are sufficiently well-described for competent researchers to confidently reproduce the main results.

**Q3 Main Strengths:**

The main idea, representing the problem as a CSP and solving it in its dual form, is original and promising. The complexity of the approach is established

**Q4 Main Weakness:**

The paper is often unclear in its writing, and includes an impressionnant number of acronyms

Contraint programming is presented as a way to solve the problem, but the reference to the state of the art algorithms provided by the CP community  is weak; it seems to me that the algorithm implemented are a brute force algorithm and two ad hoc approaches based on dynamic programming. Being better than a brute force algorithm is not difficult :).

The  temporal and probably the spatial complexity of the approach seems to be too high (exponential in k * m, m being the number of ressources) ;  then the experiments are limited to very small values of k and m (m \geq 4, k \geq 3)

There is no comparison to any other algorithm proposed by the literature to solve the same problem (be it based on CSP or not).

**Q5 Detailed Comments To The Authors:**

* please define clearly  all the acronyms (eg "DP" - the definition is hidden in a proof) and if possible limit their number
* a sketch of the main DP algorithm could improve the paper
* what about the spatial complexity of the algorithm ?
* please have a look to the literature of constraint programming that is far from being limited to brute force algorithms nor to simple dynamic programming ones

**Q9 Complying With Reviewing Instructions:**

Yes

---

> ### Author Rebuttal · Authors · 2024-04-07
>
> Thank you so much for your comments and questions.
>
> **Why did we compare with brute-force only?**
>
> Initially, when researching k-DCGs, we were unaware of any algorithms for general k-DCGs (other than the brute-force algorithm we compared in the experiments). Even the hardness of determining the existence of a pure-strategy Nash equilibrium (PSNE) when agents have binary demand vectors was an open problem (which we showed is NP-complete in this paper). To complement the hardness results, we have been exploring potential ideas that can lead to better algorithms theoretically than the only brute-force algorithm in the game theory literature for k-DCGs.
>
> In addition, the motivation for our experiments is to alleviate any concerns about the asymptotic running times of our DP algorithms, not to make a comparative study.
>
> **Why didn’t we implement existing CSP algorithms?**
>
> You are correct in the observation that Procedure 2 of our CSP formulation opens up a range of possibilities for CSP-based search algorithms. Since this paper is focused more on the theoretical foundation, we were not able to fully explore these algorithms. As mentioned in Conclusion, we are planning to devise backtracking, backjumping, and learning algorithms for congestion games and make a comprehensive experimental study on the effectiveness of these algorithms.
>
> **Literature review:**
>
> Thanks for the suggestion. We have added the following at the end of the “Experiments” section on pg. 7.
>
> Most importantly, Procedure 2 opens up a range of possibilities for new CSP-based search algorithms rooted in, for example, backjumping and learning [Kumar, 1992, Dechter, 2003, Van Beek, 2006, Rossi et al., 2008]. We leave a comprehensive experimental study as future work.
>
> **A sketch of the main DP algorithm could improve the paper**
>
> We apologize for not providing the pseudocode for our DP algorithm. We thought it would be sufficient to explain the algorithms in detail in our text. Due to the lack of space, we have updated the Appendix to include the pseudocode (Algorithm 1) for Procedures 1 and 2, which is highlighted on pg. 13. The pseudocode for learning dynamics was already provided in Algorithm 2 (pg. 18).
>
> **What about the spatial complexity of the algorithm?**
>
> Without considering the input size, for the main DP algorithm (Theorem 5), the space complexity is $O((w_{max})^{km})$ because of the size of the table $T_i$ and we only need $T_{i-1}$ to compute $T_{i}$ (i.e., there is no need to store the other tables). For other DP algorithms, their space complexities are no worse than this one.  Please let us know if we should add this to the paper.
>
> **Anonymous link to the updated paper and Appendix:** <https://www.dropbox.com/scl/fi/a1fdoos1z5edijtusb9u8/503_updated.pdf?rlkey=f10vwxd8ale1967i8kgel0osz&dl=0>
>
> We hope we have addressed your questions adequately. If there are any further questions, we'll be happy to answer those.

---

### Official Review · Reviewer_f4mV · 2024-03-20

**Q2-1 Originality-Novelty:** 3
**Q2-2 Correctness-Technical Quality:** 3
**Q2-5 Clarity Of Writing:** 4

**Q1 Summary And Contributions:**

The paper studies the problem of computing pure-strategy Nash equilibrium (PSNE) in k-dimensional congestion games, i.e. non-cooperative games in which players select from a set of resources and the sum of demands of the players choosing a specific resource determines its cost. The paper contributes both to the theory and practice of computing PSNEs in multi-dimensional congestion games. The main contributions are a CSP-based framework for computing PSNEs for general cost functions, and learning dynamics-based PSNE computation algorithms for linear and exponential cost functions.

In addition to theoretical results on the complexity of PSNE computation, the paper also provides an implementation of the CSP-based approach and demonstrates that the asymptotic running times predicted by the theoretical analysis overestimate the running time of an actual implementation of the procedures.

**Q2-3 Extent To Which Claims Are Supported By Evidence:**

4: Excellent: all claims are supported by very convincing evidence (in the form of comprehensive experimental evaluation, rigorous mathematical proofs, detailed (pseudo-)code, precise references, well-motivated and realistic assumptions) and the authors deliver what they promise.

**Q2-4 Reproducibility:**

3: Good: key resources (e.g. proofs, code, data) are available and key details (e.g. proofs, experimental setup) are sufficiently well-described for competent researchers to confidently reproduce the main results.

**Q3 Main Strengths:**

Novell CSP-formulation of a well-motivated problem.
New theoretical results on the computation of PSNEs.
Many contributions to a not so well understood problem.
CSP-based formulation could open new lines of research into using more advanced CSP techniques for computing PSNEs.

**Q4 Main Weakness:**

Dense presentation that covers a lot of variants. Given the good running time of the DP algorithm I believe the paper could motivate the detailed analysis of the variants better.

A lot of the details of the setup are in the appendix, even details that would be required for e.g. reproducing the experiments.

**Q5 Detailed Comments To The Authors:**

First, I want to state that my own research is focused on constraint decision and optimization methods, not congestion games or theoretical analysis of congestion games. I found the CSP based approach to the problem interesting. As also mentioned in the paper’s conclusions, I see potential in further research into using advanced CSP methods for computing PSNEs even more efficiently.

The main thing I would ask the authors to expand on, relates to the potential practical significance of the analysis of the different variants of general k-DCG:s.
The procedure proposed in the paper seems to already perform quite well in actual computation of PSNEs in the general case, and could possibly improve even further with other CSP solving techniques. Furthermore, if I understand theorems 6-9 correctly, the results seem to follow from small modifications to procedure 2 or preprocessing steps. As such I was wondering what consequences theorems 6-9 might have? Do the authors see potential in developing other kind of algorithms? Do the improved theoretical bounds translate into faster algorithms in practice? Did you try experimenting with some of the variants of general DCGs outline in the paper and if so, could you observe a better running time?

All in all, I am positively inclined toward the paper as it seems to contribute with a detailed analysis of a well-motivated problem that was previously not so well understood.  However, as I am not an expert in the field of congestion games, I hesitate to make to strong judgements at this stage. My main concern relates to the extent to which the different variants analyzed in the paper can be expected to help the development of algorithmic approaches to the problem.

**Q9 Complying With Reviewing Instructions:**

Yes

---

> ### Author Rebuttal · Authors · 2024-04-07
>
> **If I understand theorems 6-9 correctly, the results seem to follow from small modifications to procedure 2 or preprocessing steps … As such I was wondering what consequences theorems 6-9 might have? Do the authors see potential in developing other kind of algorithms?**
>
> We agree with your assessment that our main approaches for improving the worst-case running times for different variants (Theorems 6-9) are providing better algorithms for Procedure 2. For instance, we gave a new partitioning approach that exploits the structure of $k$-class congestion games (Theorems 7 and 8). We also provided a new reduction idea that maps a $k$-DCG to another $k$-DCG that yields better computational benefits (Theorem 9).
>
> These give us confidence that there is potential for developing other algorithms to address these variants, but we’ll be honest in saying that we don’t have results on other algorithms yet.
>
> **Do the improved theoretical bounds translate into faster algorithms in practice? Did you try experimenting with some of the variants of general DCGs outline in the paper and if so, could you observe a better running time?**
>
> We have experimented with $k$-DCGs with binary demand vectors (a variant). As shown in Figure 2, even for very small $n=4$, our algorithm is 8 orders of magnitude faster in practice than the asymptotic running time given by the theoretical results (please see pg. 7).  We expect our algorithms to perform better in practice for other variants as well. In fact, the motivation for our experiments is to show that our CSP-inspired algorithms run much faster in practice than what the asymptotic running time suggests.
>
> Following up on your question, we have performed new experiments on learning dynamics and included them in the "Experimental Results" subsection in the Appendix (pg. 19-20). These new experiments show that the learning dynamics-based algorithms are practically fast and scale up gracefully with the numbers of agents and resources. We apologize for not doing it earlier.
>
> **Anonymous link to the updated paper and Appendix:** <https://www.dropbox.com/scl/fi/a1fdoos1z5edijtusb9u8/503_updated.pdf?rlkey=f10vwxd8ale1967i8kgel0osz&dl=0>
>
> We hope we have addressed your questions adequately. If there are any further questions, we'll be happy to answer those.

---

### Official Review · Reviewer_pjRt · 2024-03-23

**Q2-1 Originality-Novelty:** 3
**Q2-2 Correctness-Technical Quality:** 3
**Q2-5 Clarity Of Writing:** 4

**Q1 Summary And Contributions:**

The authors investigate the computation of pure-strategy Nash equilibrium (PSNE) in k-dimensional congestion games (k-DCGs). This is achieved via a CSP characterization and handling of the problem as well as with a learning-dynamics based algorithm. Various complexity results are presented along with a small set of experimental results.

**Q2-3 Extent To Which Claims Are Supported By Evidence:**

3: Good: the main claims are supported by convincing evidence (in the form of adequate experimental evaluation, proofs, (pseudo-)code, references, assumptions).

**Q2-4 Reproducibility:**

4: Excellent: key resources (e.g. proofs, code, data) are available and key details (e.g. proof sketches, experimental setup) are comprehensively described for competent researchers to confidently and easily reproduce the main results.

**Q3 Main Strengths:**

The paper considers a quite interesting problem of both theoretical and practical relevance, aspects which the authors were able to balance well. A formal and comprehensive treatment of the problem is presented both in the main body of the paper as well as in further supplementary material.

**Q4 Main Weakness:**

While I appreciate that congestion games is a special class of games in itself, one that I admittedly only have a cursory knowledge of, I was left wondering if any of the results presented in the paper could not have been similarly achieved by a reduction to normal form games and the analysis of known results in equilibria computation for that class.

**Q5 Detailed Comments To The Authors:**

Is the focus purely on pure strategies, for which NE is not guaranteed to exist in various cases, a limitation imposed by the choice of solving it via CSP? Isn't allowing mix strategies not particularly relevant? There were no comments on that.

Wouldn't correlated equilibria (CE) be a more interesting problem to consider due to the very nature of the applications modeled by congestion games, some of which cited in the paper? It seems to me that, if one would want to coordinate resource allocation among many players, that would be a better solution. Furthermore, at least for normal form games, CE are easier to compute. Also, wouldn't there be a parallel between signals in CE, which map to individual strategies for each player, and the configurations and decoupling of strategies used in the dual CSP?

Minor typos/ suggested changes:

Page 4, Proof Sketch: "verifying that a PSNE takes polynomial time" -> "verifying that a strategy profile is a PSNE takes polynomial time"? "verifying a PSNE takes polynomial time"?

Page 4, Section 4: "all player appear" -> "all players appear"

Appendix page 13: issues with references to equations.

**Q9 Complying With Reviewing Instructions:**

Yes

---

> ### Author Rebuttal · Authors · 2024-04-07
>
> Thank you so much for your comments and questions.
>
> **Reducing to normal form games:**
>
> This is a very interesting point and hits the core of *computational* game theory.
>
> The short answer is the input size (or representation size). The input size of a general k-DCG is $O(nk + n2^m)$ (i.e., for each of the n players, there is a $k$-dimensional demand vector and a strategy set of size $O(2^m)$. If we reduce it to a normal-form game, the input size will become doubly exponential: $O(n (2^m)^n)$. This is because each player has a strategy set of size $O(2^m)$. So, representing the utility of a player explicitly in normal form takes $O((2^m)^n)$ real numbers. Note that, in the congestion game literature, it is common to implicitly represent the utility of each player using a cost function.  We also note that $2^m$ is the worst-case scenario --- there are other types of commonly studied congestion games that have smaller strategy space (e.g., singleton congestion games).
>
> As a result, whenever we apply any algorithm for normal-form games to congestion games (reduced to normal form), it is bound to have a doubly exponential running time. For starters, the algorithm will have to spend $O(n (2^m)^n)$ time just to take the input.
>
> **Focus purely on pure strategies:**
>
> There are two main reasons that we focus on pure strategies.
>
> The first reason is that the congestion game literature has primarily focused on the existence of/computing a pure-strategy NE (PSNE). For congestion games with at least one PSNE, computing a PSNE is already PLS-complete. Therefore, we want to first provide a complete picture for determining or computing the existence of a PSNE in many natural variants of k-DCGs (which were not known previously).
>
> The second reason is that mixed-strategy NE computation in congestion games is the first case of a natural complete problem that’s PLS ∩ PPAD-complete [Babichenko and Rubinstein, 2021]. This is an extremely hard class of problems for computing mixed-strategy NE. The result is applicable to general k-DCGs.
>
> **Correlated equilibria (CE):**
>
> This is a great suggestion and potential future direction. Although polynomial-time algorithms for correlated equilibria are known for unweighted congestion games [Papadimitriou and Roughgraden, 2008] and there are results on the price of anarchy and price of stability for CE for one-dimensional weighted congestion games [Christodoulou et al., 2005, 2019], CE computation for general k-DCGs remains open and needs further research.
>
> **Typos:**  All corrected. Thanks!
>
> **Anonymous link to the updated paper and Appendix:** <https://www.dropbox.com/scl/fi/a1fdoos1z5edijtusb9u8/503_updated.pdf?rlkey=f10vwxd8ale1967i8kgel0osz&dl=0>
>
> We hope we have addressed your questions adequately. If there are any further questions, we'll be happy to answer them.

---

### Official Review · Reviewer_Gyuz · 2024-03-24

**Q2-1 Originality-Novelty:** 3
**Q2-2 Correctness-Technical Quality:** 3
**Q2-5 Clarity Of Writing:** 3

**Q1 Summary And Contributions:**

The paper is motivated by the challenge of computing Pure Strategy Nash Equilibria (PSNE) in multidimensional congestion games (k-DCGs), a problem of significant theoretical and practical importance in fields such as traffic management, network design, and distributed systems. The key contributions of the paper include:

Computational Complexity Analysis: Demonstrating that deciding the existence of PSNE in k-DCGs is NP-complete, even under restricted conditions such as binary and unit demand vectors. This establishes a foundational understanding of the computational challenges inherent in these games.

CSP-Inspired Algorithmic Framework: Developing a novel algorithmic framework inspired by constraint satisfaction problems (CSPs) to compute PSNE in k-DCGs with general cost functions. This approach yields polynomial-time algorithms under certain conditions and offers exponential savings compared to standard CSP algorithms.

Learning Dynamics-Based Algorithms: Introducing algorithms based on learning dynamics for computing PSNE in k-DCGs with linear and exponential cost functions. These algorithms operate in polynomial time under specific assumptions and extend to approximate PSNE computation for general cost functions.

Efficient Algorithms for Structured Costs and Demands: Proposing polynomial-time algorithms for k-DCGs with structured demands and cost functions, demonstrating that specific game structures can significantly simplify PSNE computation.

**Q2-3 Extent To Which Claims Are Supported By Evidence:**

3: Good: the main claims are supported by convincing evidence (in the form of adequate experimental evaluation, proofs, (pseudo-)code, references, assumptions).

**Q2-4 Reproducibility:**

2: Fair: key resources (e.g. proofs, code, data) are unavailable but key details (e.g. proof sketches, experimental setup) are sufficiently well-described for an expert to confidently reproduce the main results.

**Q3 Main Strengths:**

1. Introduces groundbreaking CSP-inspired and learning dynamics-based algorithms for efficiently solving PSNE in k-DCGs, addressing both general and specific cost functions.
2. Offers a thorough computational complexity analysis alongside practical solutions,

**Q4 Main Weakness:**

A weakness in the work is the absence of experimental results for the Learning Dynamics Approach and the methodologies proposed in Chapter 6. This omission limits the empirical validation of these approaches, making it challenging to assess their effectiveness and efficiency in practical scenarios.

**Q5 Detailed Comments To The Authors:**

The explanation of Congestion Games was very clear and well-articulated. However, I found it somewhat challenging to understand the rationale behind presenting multiple approaches for addressing the computation of PSNE in k-DCGs. My understanding is that the selection of an appropriate algorithm and its computational complexity varies depending on the cost functions and demand functions in k-DCGs, hence the need to define solutions specific to each problem type. Is this understanding correct? If so, I believe the paper could benefit from a more explicit explanation of this reasoning within the text.

Additionally, it seems that the problem classes addressed by solutions other than the CSP approach represent a narrower class of problems when compared to CSP. Is my understanding accurate? If true, could placing CSP algorithms and those for specific classes side by side potentially confuse readers regarding the scope of applicability of each approach?

Regarding the experiments conducted with the CSP approach: Is it feasible to apply the Learning Dynamics Approach to these same problem sets? (This question is not intended to suggest adding experiments, but rather stems from an interest in the feasibility of executing the algorithm.) If feasible, could you provide insights into the expected outcomes of such experiments? Moreover, are there any algorithms currently available that could be experimentally compared with your proposed solutions?

Overall, I would appreciate if references to the Appendix within the main text could specify the exact location (e.g., section number) in the Appendix where the referenced information can be found. This would enhance navigability and understanding for readers seeking to delve deeper into the supplementary materials.

**Q9 Complying With Reviewing Instructions:**

Yes

---

> ### Author Rebuttal · Authors · 2024-04-07
>
> Thank you so much for your comments and questions.
>
> **New experimental results**
>
> We apologize for not performing experiments on learning dynamics before, thinking that the pseudopolynomial running time of the algorithms hits the point home. However, following your suggestion, we have performed new experiments and included them in the "Experimental Results" subsection in the Appendix (pg. 19-20).
>
> *Summary for your convenience:* We vary the number of dimensions $k = 2, 3, 4$ and the number of links from 2 to 10. We also vary the number of agents from 5 to 100. Figure 4 illustrates our experimental results for a subset of representative experiments. It shows that Algorithm 2 scales up gracefully as we increase the number of agents and links. This is perhaps not surprising given the pseudopolynomial running time of the algorithm. Our experiments are consistent with those on single-dimensional weighted congestion games [Panagopoulou and Spirakis, 2007].
>
> **Anonymous link to the updated paper and Appendix:** <https://www.dropbox.com/scl/fi/a1fdoos1z5edijtusb9u8/503_updated.pdf?rlkey=f10vwxd8ale1967i8kgel0osz&dl=0>
>
> **Rationale behind presenting multiple approaches**
>
> The main reason we consider the CSP-inspired and learning dynamics approaches is whether or not the existence of a PSNE is guaranteed for different classes of k-DCGs. Recall that determining the existence of a PSNE is NP-complete in general. Therefore, applying learning dynamics to general classes of k-DCGs might not lead to convergence to a PSNE (and, thus, learning dynamics would not have any computational guarantee). Instead, we only consider learning dynamics for k-DCGs that have at least one PSNE (e.g., k-DCGs with linear and exponential cost functions). For general k-DCGs (for which a PNSE might not exist), we consider the CSP-inspired approach to determine the existence of a PSNE.
>
> Therefore, yes, your understanding is correct. Following your suggestion, we have updated the paper (please see the highlighted text on pg. 2).
>
> **The problem classes addressed:**
>
> You are correct. As mentioned above, k-DCGs with linear or exponential cost functions guarantee the existence of a PSNE, whereas general k-DCGs may not have any PSNE. Therefore, our two main approaches are applied to two different classes of problems: general and restricted.
>
> **Feasibility of learning dynamics approach to these same problem sets:**
>
> Unfortunately, the learning dynamics approach does not guarantee convergence for k-DCGs and many of their variants. As a result, the learning dynamics approach is not feasible for these general problems (i.e., it may not terminate at all).
>
> **Are there any algorithms currently available that could be experimentally compared with your proposed solutions?**
>
> We are unaware of any algorithms for general k-DCGs (other than the brute-force algorithm we compared in the experiments). Within the congestion games literature spanning 50 years, no prior work has explored the proposed CSP-inspired approach or other alternative approaches for determining the existence of a PSNE for general k-DCGs.
>
>
> **Referencing Appendix**
>
> The section numbers in the appendix correspond exactly to the section numbers in the main paper. In fact, we have a single LaTeX file for the main body and appendix all together. We used a special “appendix-only” macro to designate the content of the appendix.
>
> We hope we have addressed your questions adequately. If there are any further questions, we'll be happy to answer them.

---

### Official Review · Reviewer_1BGN · 2024-03-25

**Q2-1 Originality-Novelty:** 3
**Q2-2 Correctness-Technical Quality:** 3
**Q2-5 Clarity Of Writing:** 2

**Q1 Summary And Contributions:**

This paper addresses the problem of computing Nash equilibria on pure strategies in weighted congestion gameswhen players have demand vectors. For this, two computational approaches are considered, namely constraintsatisfaction problems (CSPs) and learning dynamics, to propose algorithms for computing PSNEs.

**Q2-3 Extent To Which Claims Are Supported By Evidence:**

3: Good: the main claims are supported by convincing evidence (in the form of adequate experimental evaluation, proofs, (pseudo-)code, references, assumptions).

**Q2-4 Reproducibility:**

3: Good: key resources (e.g. proofs, code, data) are available and key details (e.g. proofs, experimental setup) are sufficiently well-described for competent researchers to confidently reproduce the main results.

**Q3 Main Strengths:**

This is a good attempt to find solutions for computing Nash equilibrium under pure straties with congestion games.This paper did mentioned the ideas of algorithms to solve the problem but no pseudo-code was provided.

**Q4 Main Weakness:**

The novelty of the ideas behind this paper is not very clear. Both of computational approaches to compute PSNEsalready proposed many years ago with different implementations. The proposed model that called k-dimensionalcongestion game in the rest of paper, is the classic weighted congestion game. Some notations never linked to themodel like p that was determined as the maximum number of strategies for players.

**Q5 Detailed Comments To The Authors:**

1. What is the reason for considering two computational approaches, namely, constraint satisfaction problems (CSPs) and learning dynamics for computing PSNEs?
2. What is the utility of the players in k-DCGs?
3. You have attached R code. Is there other representation of your algorithms, say using pseudo codes

**Q9 Complying With Reviewing Instructions:**

Yes

---

> ### Author Rebuttal · Authors · 2024-04-07
>
> Thank you for the review and helpful suggestions.
>
> **Novelty:**
> We would like to clarify the novelty of our work.
>
> First, for binary vector k-DCGs, it is an open problem whether determining the existence of a PSNE is NP-complete. This paper addresses this question by showing it is NP-complete for binary demand vectors for different sizes of k (see Theorems 3-4) via new reductions.
>
> Second, for determining the existence of a PSNE in general k-DCGs (and other subclasses), we introduce a CSP-inspired approach. However, different from the traditional CSP approach, our main innovations are: (1) decoupling the agents’ strategies and (2) computing the domains of some but not all dual variables. These save computation time significantly. Using the domains, we additionally provide more efficient algorithms to search for a PSNE (through Procedure 2) rather than using traditional off-the-shelf CSP algorithms which are more computationally expensive asymptomatically. We also discuss this after Theorem 5. Within the congestion games literature spanning 50 years, no prior work has explored the proposed CSP-inspired approach. The richness and vastness of the literature speak to the novelty of our approach.
>
> Third, when considering k-DCGs with at least one PSNE (i.e., with linear and exponential cost functions), we apply the traditional well-studied learning dynamics approach that runs natively on these k-DCGs. However, to derive the runtime of learning dynamics, we present argument and analysis through deriving potential functions explicitly to bound the runtime.
>
> Fourth, through our careful analysis, the results (in Section 6) for the structured costs and demands for k-DCGs are completely new.
>
> **No pseudo-code was provided:**
>
> We apologize for not providing the pseudocode for our algorithms. We thought it would be sufficient to explain the algorithms in detail in our text. Due to the lack of space, we have updated the Appendix to include the pseudocode (Algorithm 1) for Procedures 1 and 2, which is highlighted on pg. 13. The pseudocode for learning dynamics was already provided by Algorithm 2 (pg. 18).
>
> **Anonymous link to the updated paper and Appendix:** <https://www.dropbox.com/scl/fi/a1fdoos1z5edijtusb9u8/503_updated.pdf?rlkey=f10vwxd8ale1967i8kgel0osz&dl=0>
>
> **Q1 What is the reason for considering two computational approaches, namely, constraint satisfaction problems (CSPs) and learning dynamics for computing PSNEs?**
>
> The main reason we consider the CSP-inspired and learning dynamics approaches is whether or not the existence of a PSNE is guaranteed for different classes of k-DCGs. Recall that determining the existence of a PSNE is NP-complete in general. Therefore, applying learning dynamics to general classes of k-DCGs might not lead to convergence to a PSNE (and, thus, learning dynamics would not have any computational guarantee). Instead, we only consider learning dynamics for k-DCGs that have at least one PSNE (e.g., k-DCGs with linear and exponential cost functions). For general k-DCGs (for which a PNSE might not exist), we consider the CSP-inspired approach to determine the existence of a PSNE.
>
> We have updated the paper to explicitly mention this (please see the highlighted text on pg. 2).
>
> **Q2 What is the utility of the players in k-DCGs?**
>
> The utility is the negative of the cost function denoted by $\pi_i({\mathbf{s}})$ in Section 2. Notably, we define PSNE in Definition 1 using the cost function $\pi_i({\mathbf{s}})$, which is standard in the congestion game literature.
>
> **Q3 You have attached R code. Is there other representation of your algorithms, say using pseudo codes**
>
> We have updated the paper to address this, as mentioned above.
>
> We hope we have addressed your questions adequately. If there are any further questions, we'll be happy to answer them.

---

### Meta-Review · Area_Chair_AKpN · 2024-04-17

The paper gives a thorough computational complexity analysis alongside practical solutions for multidimensional congestion games.
I am less convinced by the very basic CSP approach they proposed. It would be interesting to test CP solvers (e.g. Minizinc solvers) with rich global constraints and better dynamic programming algorithms exploiting the problem structure (i.e., AND/OR search, BTD,..). This would require further work, possibly outside the scope of this more theoretical paper.